# The impacts of pollution sources and temperature on the light absorption of HULIS were revealed by UHPLC-HRMS/MS at the molecular structure level

*Tao Qiu[a], Yanting Qiu[b], Yongyi Yuan[a], Rui Su[c], Xiangxinyue Meng[b], Jialiang Ma[d], Xiaofan Wang[b], Yu Gu[a], Zhijun Wu[b,]\*, Yang Ning[a], Xiuyi Hua[a], Dapeng Liang[a,]\*, Deming Dong[a]*

[a] Key Lab of Groundwater Resources and Environment of the Ministry of Education, College of New Energy and Environment, Jilin University, Changchun 130012, China

[b] State Key Laboratory of Regional Environment and Sustainability, College of Environmental Sciences and Engineering, Peking University, Beijing 100871, China

[c] State Key Laboratory of Inorganic Synthesis and Preparative Chemistry, College of Chemistry, Jilin University, Changchun 130012, China

[d] Department of Chemistry, Aarhus University, 8000 Aarhus C, Denmark

*Corresponding Author: Dapeng Liang (liangdp@jlu.edu.cn) and Zhijun Wu (zhijunwu@pku.edu.cn)

**ABSTRACT.** Atmospheric humic-like substances (HULIS), a key component of brown carbon (BrC), significantly promote the light absorption of aerosols. However, their linkages to pollution sources and ambient temperature in cold environments remain unresolved. Here, we analyze wintertime urban aerosol samples in Changchun, Northeast China, using ultrahigh performance liquid chromatography coupled with high-resolution tandem mass spectrometry (UHPLC-HRMS/MS). HULIS show a high light absorption efficiency ($MAE_{365}$ = 1.81 ± 0.24 $m^2$ $gC^{-1}$) and high mass concentration (2.97 ± 1.54 $\mu$gC $m^{-3}$), exceeding values reported from other global regions. Through UHPLC-HRMS/MS characterization, we identify 264 compounds at the molecular structure level, accounting for 38.2 - 78.1% of the total HULIS mass. Compositional analysis demonstrates biomass burning and coal combustion are the main BrC sources during haze events. We screen out 39 strong BrC chromophores, mainly nitrophenols, that contribute 28.9 ± 10.4% of the total light absorbance at 365 nm. Low ambient temperatures potentially enhance the accumulation of these strong BrC chromophores in the aerosol particles by suppressing photobleaching processes and altering thermodynamic reaction equilibria. These findings emphasize the potential of BrC to exert a more significant and persistent environmental effect in the cold region atmosphere.

**KEYWORDS:** *Humic-Like Substances, Non-targeted screening, Northeast China, Light absorption efficiency, Brown carbon chromophores.*

**Graphical Abstract**

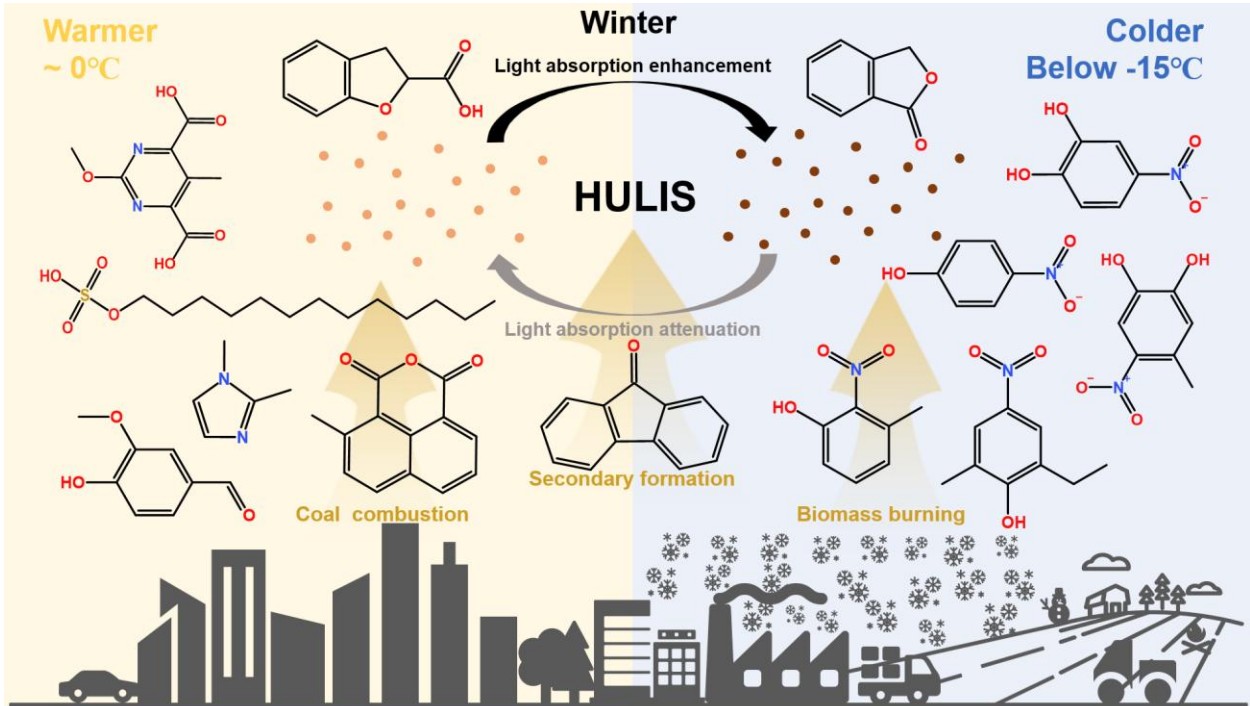

**1. Introduction**

32        Atmospheric humic-like substances (HULIS) are important components in light-absorbing aerosols (Hoffer et al.,
2006; Zou et al., 2023), therefore affecting global radiative forcing and atmospheric chemical processes (Chung et al.,
2012; Huang et al., 2020; Laskin et al., 2015). They are identified as a highly complex aggregate of polar organic
compounds composed of aromatic, aliphatic, and alicyclic structures with functional groups such as hydroxy, carbonyl,
carboxyl, nitrooxy, and sulfooxy (Song et al., 2018; Wang et al., 2019; Zou et al., 2023). Previous studies have revealed
that the molecular composition of HULIS determine their physicochemical properties, further impacting their climatic
and environmental effects, such as cloud condensation nuclei activation, human health, and global radiation (Bao et
al., 2023; Cappiello et al., 2003; Chen et al., 2021; Dou et al., 2015; Hems and Abbatt, 2018; Krivácsy et al., 2000).

40        As reactive components in the atmosphere, HULIS exhibit pronounced chemical activity through their
oxygenated functional groups, particularly prone to the oxidation by reactive oxygen radicals and gaseous oxidants
(Hems et al., 2021; Huo et al., 2021; Qiu et al., 2024). Both laboratory simulations and field observations have
demonstrated that these atmospheric aging processes significantly alter the light-absorption properties and
environmental behaviors of HULIS (Hems and Abbatt, 2018; Qiu et al., 2024; Wang et al., 2022, 2019). Furthermore,
significant efforts have been directed towards understanding the link between molecular composition and light
absorption of HULIS. Studies have suggested that chromophores like nitroaromatics and oxygenated polycyclic
aromatics are key contributors to the light absorption of HULIS (Kuang et al., 2023; Qin et al., 2022; Song et al., 2019;
Zou et al., 2023). However, critical knowledge gaps persist regarding the molecular structures that dominate light
absorption and, importantly how these molecules and their associated absorption properties evolve during atmospheric
aging processes. This limits comprehension understanding of the atmospheric evolution process and radiative effect
of HULIS.
Once emits into or form in the atmosphere, vertical transport increased the altitude of HULIS-containing particles,
leading to long-range transport (Chen et al., 2021; Slade et al., 2017). During vertical transportation, the ambient
temperature sharply decreases, indicating that the atmospheric evolution of HULIS accompanies with low-temperature
conditions during their majority lifetime (Heald et al., 2005; Liu et al., 2014; Pani et al., 2022; Textor et al., 2006; Wu
et al., 2018a). How low temperature impacts the atmospheric evolution process of HULIS remains uncertain yet. The
decrease in temperature could potentially alter the physicochemical properties of HULIS, influencing their volatility
(Cao et al., 2018; Schervish and Donahue, 2020), reactivity (Liu et al., 2023; Slade et al., 2017), and partitioning
between the gas and particulate phase (Arp et al., 2008; Tao and Murphy, 2021). Consequently, their light absorption
and atmospheric lifetime are profoundly affected (Gregson et al., 2023; Roelofs, 2013). Therefore, it is imperative to
further explore the low-temperature behavior of HULIS through field observations in cold environments.
This study collected atmospheric $PM_{2.5}$ samples in Changchun, which experiences low temperatures in
wintertime, and HULIS were subsequently extracted. By employing a combination of non-targeted analysis and
ultrahigh performance liquid chromatography coupled with high-resolution tandem mass spectrometry (UHPLC-
HRMS/MS), we aimed to identify the molecular structures in HULIS collected from urban aerosols during wintertime.
This approach provided us new insights into the potential sources and temperature effects on the light-absorption
properties of HULIS.

## 68  2. Experimental Section

**2.1 Aerosol sampling and HULIS extraction.** We conducted a field campaign on the campus of Jilin University
in Changchun, Northeast China (125.29° E, 43.83° N) from January 1[st] to 30[th], 2023. During this period, a high-
volume particulate sampler (Tianhong Intelligent Instrument Plant, Wuhan, China, 1.05 $m^3$ $min^{-1}$) collected 24 h $PM_{2.5}$
samples on a pre-baked quartz filter. **Figure S5** presented the meteorological data (http://www.wunderground.com/)
and air pollutants data (http://air.cnemc.cn:18007/), including relative humidity, temperature, concentrations of CO,
$SO_2$, $O_3$, $NO_2$, $PM_{2.5}$, and $PM_{10}$ during the sampling campaign.
The preparation process of HULIS sample was identical with previous studies (Limbeck et al., 2005; Yuan et al.,
2021; Zou et al., 2020), and can be briefly described as the following steps: sampled filter was firstly extracted with
ultrapure water (> 18.2 MΩ) in an ultrasonic bath for 40 mins. After that, the water extracts were filtered through 0.22
$\mu$m PES syringe filters, then acidified to pH=2 by HCl solution (0.1 M) and loaded on the pre-acidification solid phase
extraction (SPE) cartridges (Supelclean ENVI-18, 500 mg, 3 mL). The majority of inorganic ions, low molecular-
weight organic acids, and sugars were eluted out with ultrapure water while the fractions retained in the SPE cartridge
were eluted with methanol (Baduel et al., 2009). Finally, a portion of the elution was measured by UHPLC-HRMS/MS
and the rest was dried under a gentle stream of $N_2$, then redissolved in ultrapure water for total organic carbon and
light absorption analysis.
**2.2 Molecular composition analysis of HULIS.** An ultrahigh performance liquid chromatography system
(UHPLC, Dionex Ultimate 3000, Thermo Fisher Scientific, San Jose, CA, U.S.A.) coupled with an Orbitrap Fusion
Tribrid mass spectrometer (Thermo Fisher Scientific, San Jose, CA, U.S.A.) was used to detect the molecular
composition of HULIS. To detect as many HULIS species as possible and achieve quantification, we optimized the
detection method (detailed in **Text S1**) to decrease the method detection limit and applied a semi-quantitative strategy
to quantify the identified compounds.
The optimized chromatographic conditions were as follows: Accucore C18 2.6 $\mu$m particle size (100 × 2.1 mm,
Thermo Scientific) with the gradient elution started from 80% of mobile phase A (0.05% acetic acid) with a 0.2 mL
min$^{-1}$ flow rate for 2 min, then changed to 100% of mobile phase B (methanol with 0.05% acetic acid) in 15 min and
maintained constant for 2 min, decreased to 20% of mobile phase B within 1 min and finally held for 3 min for re-
equilibration. The mass spectra (*m/z* 60-600) with a resolving power of 120,000 (*m/z* 200) were obtained by using
heated-electrospray ionization (H-ESI). The optimized mass spectrometric parameters were as follows: 3.5 kV spray
voltage for positive ions and 3.25 kV spray voltage for negative ions, 35 psi sheath gas (nitrogen), and 10 psi auxiliary
gas (nitrogen), 320 °C ion transfer tube temperature, 125 °C vaporizer temperature. The data acquisition used data
dependent mode and the master scans interval time was set as 1.0 second for the full scan experiments (detailed in
**Table S5**).
The obtained data analysis was performed with the Compound Discoverer 3.3 software to generate
reasonable molecular formulas and match fine structures to MS/MS data. The numbers of atoms restriction of
formula were 1-40 for C, 1-100 for H, 0-40 for O, 0-6 for N, and 0-2 for S, with $0.3 \leq H/C \leq 3.0$, $0 \leq O/C \leq 1.2$,
$0 \leq N/C \leq 1.0$, and $0 \leq S/C \leq 0.8$ (Kind and Fiehn, 2007). All of the mathematically formulas for each peak were
performed with a mass tolerance of ± 5 ppm and peak areas more than three times of the blank sample. Three
curated spectral databases, mzcloud library database, ChemSpider library database, and CFM-ID
(https://cfmid.wishartlab.com) were applied to screen suspect candidates of structure (Allen et al., 2015).
According to the Schymanski's confidence level (CL), these candidates were divided into confirmed structures
(CL1), probable structures (CL2), and tentative candidates (CL3) (Schymanski et al., 2014). We showed two
examples to illustrate the derivation processes of candidates in **Figure S6**.
A semi-quantitative strategy was conducted as follows: target analytes were quantified using external
standard solutions of structurally analogous surrogate compounds (Nguyen et al., 2014; Nozière et al., 2015). A
representative application involved utilizing the standard curve of 4-methyl-5-nitrocatechol to simultaneously
quantify three structural analogs: 4-methyl-5-nitrocatechol, 3-methyl-5-nitrocatechol, and 3,4-dimethyl-5-
nitrocatechol. While this strategy enables quantification of compounds without commercially available standards,
it introduces inherent uncertainties due to ionization efficiency variations between surrogates and target analytes
(discussed in **Text S2**).
**2.3 Light absorption analysis of HULIS and other analysis.** A total of the HULIS extract was first diluted to
3 mL with ultrapure water and then measured by a UV-Vis spectrophotometer (UV-1900, Shimadzu, Kyoto, Japan) at
200-700 nm with an interval wavelength of 1 nm. To assess the optical properties of HULIS samples, the mass
absorption efficiency ($MAE_\lambda$, m$^2$ gC$^{-1}$) was calculated according to the following formula.
$$Abs_\lambda = (A_\lambda - A_{700}) \frac{V_l}{V_a \times l} \ln 10 \qquad (1)$$
$$MAE_\lambda = \frac{Abs_\lambda}{c} \qquad (2)$$
where $Abs_\lambda$ represents the light absorption coefficient of the HULIS extract at a wavelength of $\lambda$ nm (Mm$^{-1}$), $A_\lambda$ is
the recorded absorbance value of the HULIS extract by the UV-Vis spectrophotometer, $V_l$ is the total solution volume
of HULIS extract (mL), $V_a$ is the air sampling volume corresponding to the volume of HULIS extract (m$^3$), $l$
represents the optical path length (0.01 m), $C$ is the mass concentration of HULIS carbon (HULIS-C) ($\mu$gC m$^{-3}$).
The contents of elemental carbon (EC) and organic carbon (OC) in quartz fiber filters were determined by a
Thermo-Optical Transmission (TOT) method on a Sunset Lab EC/OC analyzer. The concentrations of HULIS-C were
analyzed by a total organic carbon (TOC) analysis (TOC-L, Shimadzu, Kyoto, Japan). Additionally, the ratio of OC
emitted from combustion to TOC was calculated by the assigned formula (Cabada et al., 2004).
$$OC_{com}/OC_{total} = (\left[\frac{OC}{EC}\right]_p * [EC])/[OC] \qquad (3)$$
where $OC_{com}/OC_{total}$ represents the ratio of OC emitted from combustion to total OC, $[OC/EC]_p$ is the ratio of
OC to EC for the primary sources, $[EC]$ and $[OC]$ are the measured EC and OC concentration, respectively.
Water-soluble inorganic ions in PM$_{2.5}$ collected on Teflon filter were detected by ion chromatography (IC,
Shimadzu, Kyoto, Japan). The aerosol liquid water content (ALWC) and pH were calculated using the ISORROPIA-
II thermodynamic model based on meteorological data and mass concentrations of water-soluble inorganic ions
(Fountoukis and Nenes, 2007; Nenes et al., 1998; Wu et al., 2018b).

## 3. Results and Discussion

**3.1 Molecular composition and light absorption of HULIS. Figure 1A** illustrated the temporal variations of
HULIS-C, OC, and EC mass concentrations during the sampling period. The temperatures ranged from 2.9 to -25.3 ℃
and solar radiation ranged from 24.3 to 57.8 W m$^{-2}$ (**Figure 1B**). The average concentrations of OC and EC were 11.7
$\pm$ 5.74 and 2.06 $\pm$ 0.92 $\mu$g m$^{-3}$, respectively. The average HULIS-C concentration was 2.97 $\pm$ 1.54 $\mu$g m$^{-3}$, accounting
for 25.1% of total OC. The observed HULIS-C concentration was higher than those observed in winter of Europe
(0.68 – 1.47 $\mu$g m$^{-3}$) (Emmenegger et al., 2007; Voliotis et al., 2017), South America (0.20 – 1.30 $\mu$g m$^{-3}$) (Serafeim
et al., 2023), and Chinese other regions (1.96 – 2.38 $\mu$g m$^{-3}$) (Lu et al., 2019; Ma et al., 2019; Zou et al., 2023),
indicating the abundance of HULIS in Changchun.

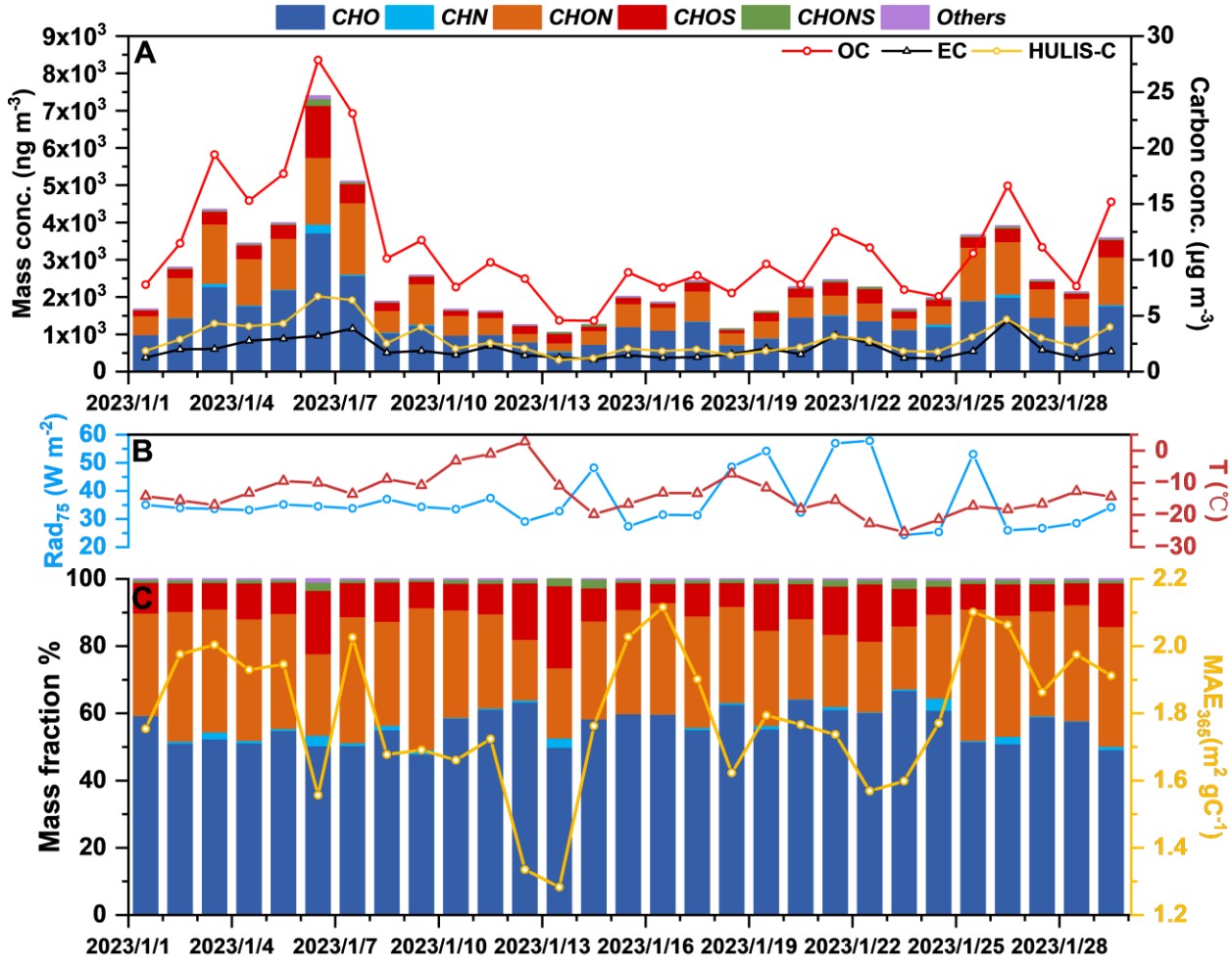

**Figure 1.** Temporal variations in the mass concentrations of six compound categories, organic carbon, and elemental carbon (A); the 75th percentile of solar radiation ($Rad_{75}$) and ambient temperature (B); the mass fraction of six compound categories as well as $MAE_{365}$ (C).

Non-targeted analysis of HULIS by UHPLC-HRMS/MS revealed 264 compounds at Schymanski's confidence levels above CL3 (Schymanski et al., 2014). **Table S7** listed the details of detected HULIS compounds and their corresponding CLs. The identified compounds were grouped into six compound categories, including CHO (composed of carbon, hydrogen, and oxygen atoms, hereinafter), CHN, CHON, CHOS, CHONS, and other species. CHONS category refers to compounds that contain carbon, hydrogen, oxygen, nitrogen, and sulfur elements. **Figure 1A and 1C** showed temporal variations of the mass concentrations and fractions of different compound categories. The total mass concentration of these compounds ranged from 1.05 to 7.39 $\mu g\ m^{-3}$ (**Figure 1A**), explaining 38.2% - 78.1% of the total HULIS mass (converted by multiplying [HULIS-C] by 1.6, (Friman et al., 2023)). The remaining unidentified compounds in HULIS mainly include low-polarity phenols, ketones, and aldehydes, which cannot be detected bythe ESI mode (Huang et al., 2025; Huo et al., 2021; Song et al., 2022, 2024).

**Figure 1C** also displayed that MAE at 365 nm ($MAE_{365}$) of HULIS samples. The observed $MAE_{365}$ ranged from 1.28 to 2.12 $m^2\ gC^{-1}$ (1.81 ± 0.24 $m^2\ gC^{-1}$ in average), which was higher than those in Beijing (1.79 ± 0.24 $m^2\ gC^{-1}$) (Cheng et al., 2011), Xi'an (1.65 ± 0.36 $m^2\ gC^{-1}$) (Huang et al., 2018), Guangzhou (1.1 ± 0.27 $m^2\ gC^{-1}$) (Zou et al.,

2023), and Hong Kong ($0.97 \pm 0.40$ $m^2$ $gC^{-1}$) (Ma et al., 2019) during wintertime, demonstrating that the light absorption efficiency of HULIS in Changchun was higher than other regions in China. Moreover, the strongly positive correlation (**Figure S7**) between $MAE_{365}$ with CHON category (Pearson's R = 0.75, *p*-value < 0.01) and aromatic fraction (Pearson's R = 0.86, *p*-value < 0.01) suggested that the high light absorption efficiency of HULIS may be related to aromatic CHON compounds.

**3.2 Potential sources of HULIS based on molecular analysis.** To investigate drivers of the high concentrations and variable light absorption efficiency of HULIS in this study, we selected two samples (Event I and II) among all haze events ($PM_{2.5}$ concentration > 75 $\mu g$ $m^{-3}$) that exhibited the maximal divergence in $MAE_{365}$ values. Event I had higher $PM_{2.5}$ ($159.6 \pm 53.8$ $\mu g$ $m^{-3}$) and HULIS-C (6.68 $\mu gC$ $m^{-3}$) but lower $MAE_{365,HULIS}$ (1.56 $m^2$ $gC^{-1}$), while Even II had lower $PM_{2.5}$ ($83.7 \pm 36.4$ $\mu g$ $m^{-3}$) and HULIS-C (4.65 $\mu gC$ $m^{-3}$) but higher $MAE_{365,HULIS}$ (2.06 $m^2$ $gC^{-1}$). These contrasting events were chosen for potential sources comparison from the perspective of molecular composition. Considering the lowest $PM_{2.5}$ and HULIS-C concentration, the sample on January 13 ($PM_{2.5}$ = $14.1 \pm 11.9$ $\mu g$ $m^{-3}$, HULIS-C = 0.97 $\mu gC$ $m^{-3}$, $MAE_{365,HULIS}$ = 1.28 $m^2$ $gC^{-1}$) was selected to represent clean days. **Figure 2** exhibited the reconstructed MS spectra, the number, and concentration fraction of HULIS in both positive and negative modes.

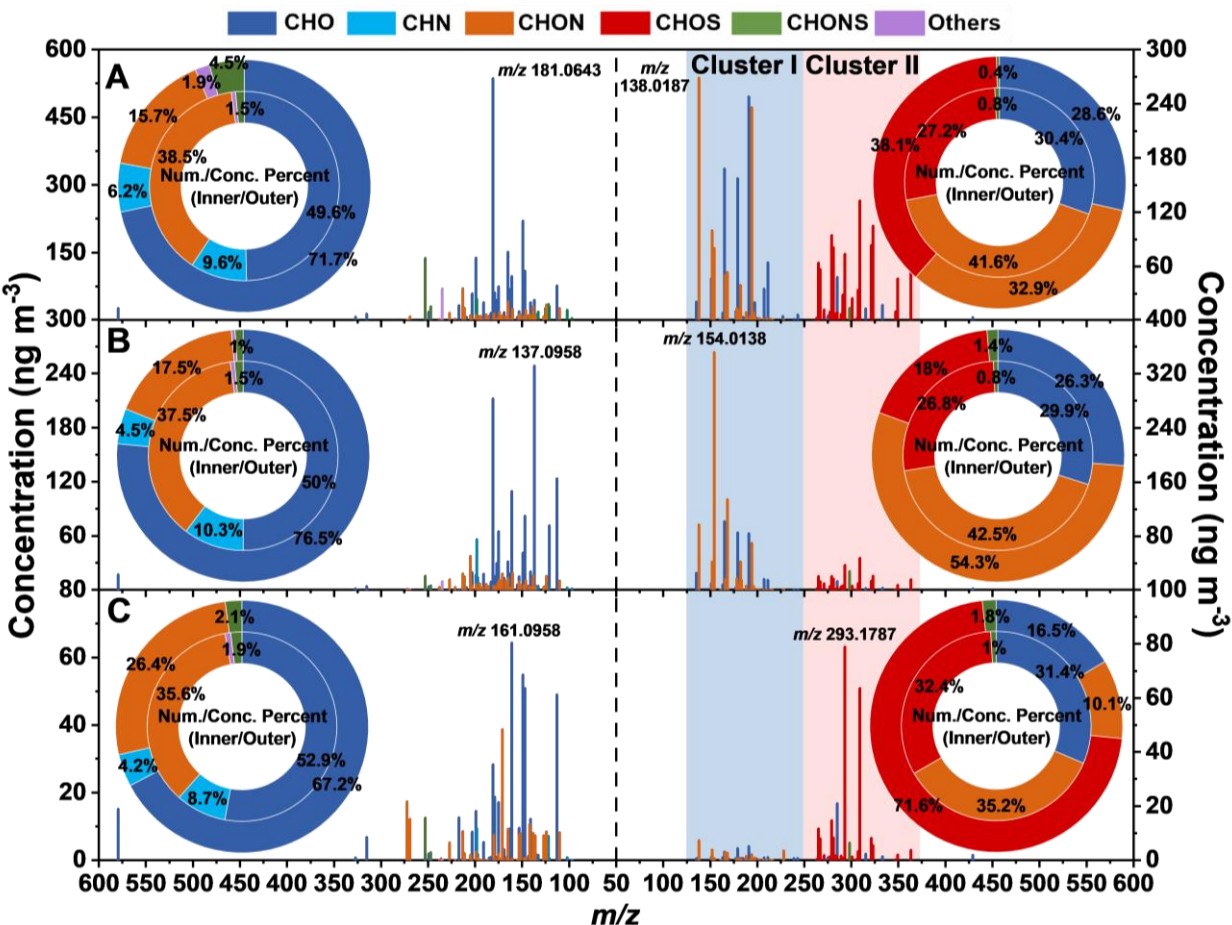

**Figure 2.** Reconstructed mass spectra of identified HULIS compounds during Event I (A), Event II (B), and Clean day (C). Spectra are shown for positive ionization mode (left panels) and negative ionization mode (right panels). *m/z* values increase from middle to both sides in all spectra. The most abundant ions are labeled with their *m/z* values. The accompanying pie charts represent the

molecular class distribution of the identified compounds: the inner/outer ring shows the relative abundance based on number/concentration of compounds.

In the positive mode, Event I and II had similar molecular composition, both dominated by CHO compounds, followed by CHON, CHN, and others. The most abundant species in Event I and II were 9-fluorenone ($m/z$ 181.0643) and 2-[(1E)-1-Buten-1-yl]-5-methylfuran ($m/z$ 137.0958), respectively. The former originates from diverse combustion sources such as biomass burning, coal combustion, and vehicle emission (Alves et al., 2016; Huo et al., 2021; Ma et al., 2023; Souza et al., 2014; Xu et al., 2024; Zhao et al., 2020), whereas the latter is believed to stem specifically from biomass burning (Bhattu et al., 2019; Hatch et al., 2015). High concentrations of biomass burning ($K^+$) and coal combustion ($SO_2$, **Table S6**) tracers proved the key contribution of biomass burning and coal combustion (Chen et al., 2017; Dutton et al., 2009; He et al., 2010; Liang et al., 2021), which have been confirmed in our previous study to be the main sources of air pollution in Changchun winter (Dong et al., 2023).

In the negative mode, two distinct compound clusters were observed within the $m/z$ range of 125 – 250 (refer to Cluster I) and 250 – 375 (refer to Cluster II), as marked in the right part of **Figure 2**. Cluster I comprised a significant proportion of strong BrC species, such as nitrophenols (including 4-nitrophenol, 3-nitrocatechol, 4-nitro-1-naphthol, and etc., **Table S7**), mainly originating from primary emissions like biomass burning and coal combustion (Huang et al., 2023; Jiang et al., 2023; Lin et al., 2017; Wang et al., 2020) and secondary formation (Bolzacchini et al., 2001; Mayorga et al., 2021). Notably, the higher abundance of Cluster I in Event II compared to Event I likely contributed to the higher $MAE_{365}$ observed during Event II.

All of CHOS compounds were characterized by ion fragment $m/z$ 96.9595 in the MS/MS spectra and were therefore identified as organosulfates (OSs). Cluster II was predominantly composed of OSs, accounting for 84.7 ± 6.5 % of compounds within this cluster during the whole sampling period. Considering the OSs are typically formed by atmospheric aqueous reaction (Brüggemann et al., 2017; Pratt et al., 2013; Wach et al., 2020), the dominance of OSs in Cluster II strongly supports its secondary formation origin. The higher abundance of Cluster II in Event I indicated more intense secondary formation of HULIS during this event compared to Event II. This interpretation is corroborated by elevated concentrations of secondary inorganic ions (including $NH_4^+$, $NO_3^-$, and $SO_4^{2-}$, 16.66 – 41.56 vs 4.30 – 7.30 $\mu$g m$^{-3}$) and relative humidity (83.1 ± 4.6% vs 61.9 ± 14.0%) in Event I vs. II, as detailed in **Table S6**. As a result, the higher ALWC (95.9 vs 23.9 $\mu$g m$^{-3}$) and lower pH value (5.29 vs 7.31) in Event I in contrast to Event II facilitated the formation of OSs. Since the OSs studied here were primarily aliphatic sulfates (summarized as the molecular formulas of $C_nH_{2n+2}O_{4-6}S$ and $C_nH_{2n}O_{4-6}S$, where $10 \le n \le 18$), which belong to non-light-absorbing organic matter, this may cause the lower $MAE_{365}$ value in Event I.

**3.3 Effect of ambient temperature on the BrC chromophores of HULIS.** As above-mentioned, the temperature was down to -25℃. This extreme cold temperature critically alters the reactivity, phase partitioning, and aging kinetics of HULIS(He et al., 2006; Huang et al., 2006; Li and Shiraiwa, 2019; Shiraiwa et al., 2011). In total, 39 compounds were screened as strong BrC chromophores to investigate the effect of temperature on the BrC chromophores according to a partial least squares regression (PLS) model (detailed in **Text S3**). These compounds belong to nitrophenols or nitrophenol derivatives, which are marked in **Table S7**. The mass concentration of these 39 strong BrC chromophores was 0.41 ± 0.27 $\mu$g m$^{-3}$ in average, accounting for 8.67 ± 3.68% of the total HULIS mass (converted from [HULIS-C] using a factor of 1.6, (Friman et al., 2023)). This mass fraction exceeds values reported for 12 specific nitro-aromatic compounds in previous studies (about 7.5% of HULIS mass, (Frka et al., 2022)). Despite

this modest mass contribution, these strong BrC chromophores contributed 28.9 ± 10.4% of the light absorbance (detailed in **Text S4**), with an average $MAE_{365}$ of 7.40 ± 1.80 $m^2$ $gC^{-1}$ (**Figure S7**), higher than 10% and 14% light absorbance contribution of 18 chromophores in Xi'an and Beijing (Huang et al., 2020).

**Figure 3** shows that the mass fraction of screened 39 strong BrC chromophores under different temperature ranges, as well as the negative variation patterns between the $MAE_{365}$ and ambient temperature. In contrast, the $OC_{com}/OC_{total}$ ratio shows no consistent temperature dependence, suggesting low temperature rather than combustion emission promote the accumulation of strong BrC species in the particles. We proposed two possible explanations: firstly, the low temperature may lead to a non-liquid state of ambient particles, potentially introducing kinetic limitation on the diffusion of reactive species from gas phase into particle bulk (Li and Shiraiwa, 2019). We utilized an established parameterization scheme (**Text S5**) to calculate the glass transition temperature (Tg) of HULIS based on their molecular composition (Li et al., 2020). The results showed that the decrease in ambient temperature (T) enhanced the Tg/T ratio, driving the phase transition of particles from liquid state (Tg/T = 0.76) to semi-solid state (Tg/T > 0.79). This may lead to the diffusion coefficients reduction of reactive species (Arangio et al., 2015; Gatzsche et al., 2017; Mikhailov et al., 2009; Shiraiwa et al., 2011; Virtanen et al., 2010), thereby slowing the degradation rate of BrC via hydroxyl radical oxidation or triplet excitation pathways in the atmosphere (Schnitzler et al., 2022; Schnitzler and Abbatt, 2018). These findings suggest that the non-liquid particle phase state, accompanied with the weak solar radiation during Changchun's winter (refer to **Figure 1B**), results in a less pronounced photochemical aging of BrC, thereby diminishing its photobleaching.

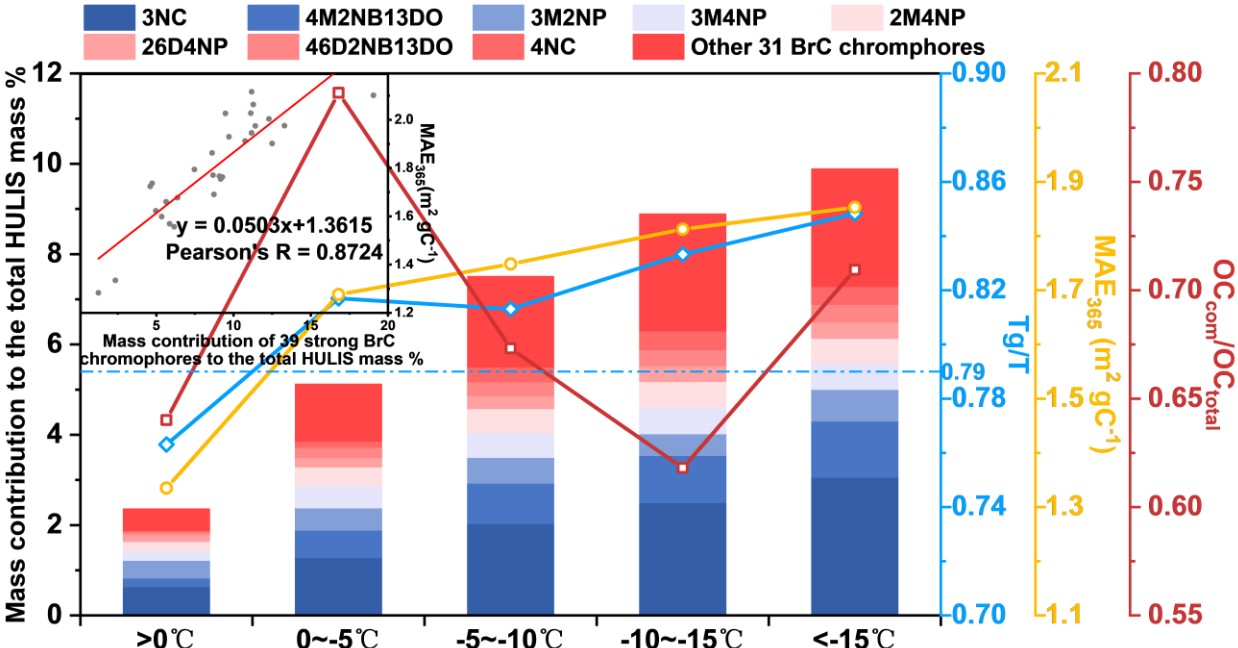

**Figure 3.** Temperature-dependent variations in mass fraction of 39 strong BrC chromophores, $MAE_{365}$ value of HULIS, Tg/T ratio, and $OC_{com}/OC_{total}$ ratio, with correlation between chromophore mass fraction and HULIS $MAE_{365}$. The blue dotted line represented the threshold of Tg/T between liquid and semi-solid state (Shiraiwa et al., 2017), and the abbreviations of 3NC, 4M2NB13DO, 3M2NP, 3M4NP, 2M4NP, 26D4NP, 46D2NB13DO, and 4NC represents 3-nitrocatechol, 4-methyl-2-nitrobenzene-1,3-diol, 3-methyl-2-nitrophenol, 3-methyl-4-nitrophenol, 2-methyl-4-nitrophenol, 2,6-dimethyl-4-nitrophenol, 4,6-dimethyl-2-nitrobenzene-

1,3-diol, and 4-nitrocatechol, respectively.

247        Secondly, the formation of BrC chromophores was also important for the $MAE_{365}$ enhancement of HULIS. On
the one hand, the secondary formation of nitrophenols has been conclusively attributed to reaction of phenols with
$NO_x$ radicals (Bolzacchini et al., 2001; Finewax et al., 2018; Kroflič et al., 2021; Mayorga et al., 2021), a process that
has been characterized as exothermic (Bolzacchini et al., 2001; Domingo et al., 2021). On the other hand, we have
demonstrated that further atmospheric oxidation of nitrophenols proceeds via a ring-opening mechanism of benzene
moiety (Qiu et al., 2024), which constitutes an endothermic reaction (Cao et al., 2021; Hems and Abbatt, 2018; Wang
et al., 2017). From a thermodynamic perspective, low temperature not only promotes exothermic formation of
nitrophenols while simultaneously suppressing their endothermic degradation via ring-opening. Furthermore, low
temperature inhibits the volatilization and enhances the particle-phase retention of these volatile chromophores (He et
al., 2006; Huang et al., 2006). This combined effect of low temperature led to the accumulation of strong BrC
chromophores like nitrophenols within HULIS. This mechanism is consistent with field observations of enhanced
nitroaromatic abundance in winter aerosols (Cai et al., 2022; Teich et al., 2017; Zhang et al., 2024). As such, we infer
that ambient temperature plays a critical role in promoting the transformation and light absorption of BrC
chromophores, particularly in cold or/and high-altitude regions.
**3.4 Conclusions.** In this work, we explored the linkage between the light absorption and molecular structure of
atmospheric HULIS in Changchun winter based on UHPLC-HRMS/MS. In different haze events, the molecular
structure of HULIS varied due to different sources, which lead to differences in their light absorption efficiency.
Biomass burning and coal combustion were important inducers of the high $MAE_{365}$ value of HULIS. It was the fact
that biomass burning and coal combustion emitted a large fraction of BrC chromophores such as nitrophenols, while
aliphatic organosulfates produced by secondary formation lead to the reduction in the light absorption efficiency of
HULIS.
In addition, our study screened out 39 strong BrC chromophores belonging to nitrophenols by PLS model. These
species accounted for 8.67 ± 3.68% of the total HULIS mass while contributed nearly 30% of the light absorbance at
365 nm. We found that low temperature can promote the accumulation of strong BrC chromophores through slowing
down the photobleaching reaction and changing the thermodynamic reaction equilibrium, thereby improving the light
absorption capability of HULIS. This phenomenon has not been studied before, and further laboratory and field studies
are urgently needed to verify the effect of temperature on the light absorption properties of BrC.
Our research has found that in the cold regions of northern China, on one hand, primary emissions from biomass
burning and coal combustion are relatively strong, and on the other hand, low temperatures reduce the photobleaching
of brown carbon (BrC). This implies that BrC in cold regions may have a longer lifetime and stronger light-absorbing
properties in the atmosphere, thus playing a more significant role in the direct radiative forcing of carbonaceous
aerosols.
**Supplementary Information.** Optimization details of LC-MS method, calculation procedure of relevant index,
screening details of PLS model, and analysis results of pollutant data and meteorological data (PDF).
The information about atmospheric mass concentration, molecular information, and strong brown carbon
chromophores of identified compounds in HULIS samples of Changchun during wintertime (Table S7).
**Author Contributions.** T. Q. and Y. Q. designed this work. T. Q., X. W., and Y. G., collected the experimental samples.
T. Q., Y. Q., Y. Y., R. S., Y. N., X. H., and X. M. collected and analyzed the experimental data. Z. W., D. L., D. D., and
J. M. edited the manuscript. All authors have read and agreed to the published version of the manuscript.
**Funding.** This work was supported by the National Natural Science Foundation of China (No. 22376084) and
Environmental Protection Research Program from Jilin Department of Ecology and Environment (No. 2023-09).
**Notes.** The authors declare no competing financial interest.
**Acknowledgments.** Authors of this article wish to thank the financial support from the National Natural Science
Foundation of China (No. 22376084) and Environmental Protection Research Program from Jilin Department of
Ecology and Environment (No. 2023-09), the technical support from Ms. Huimin Qian and the work support from Mr.
Shengao Yang. This work was also supported by State key Laboratory of Inorganic Synthesis and Preparation
Chemistry, Jilin University.

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
