# Peer review of "The impacts of pollution sources and temperature on the light"

_EGUsphere, 2025_

## Author Comment (AC1)

**Response to reviewer #3**

Thanks to the reviewer for your careful reading and your constructive comments and suggestions on our manuscript. The reviewer's comments and suggestions are shown as *italicized font*, our response to the comments is normal font. New or modified text is in normal font and in blue. Details are as follows.

**Reviewer's comments:**

Reviewer #3: *This work investigates the molecular compositions and light absorption of HULIS in cold environments. The data is informative and interesting. However, some statements need more evidence and detailed in the main text. The following comments need to be addressed before publication.*

[Response]

Thanks for the reviewer's comments on this manuscript. Please check our point-by-point response and the modified text in the manuscript.

*(1) Introduction: There have been many studies on revealing the molecular-level characteristics and compositions as well as the relation to HULIS light absorption. I may suggest the author review the literature on the HULIS molecular compositions in the introduction section. This would be helpful to gain a deeper understanding on the findings in this work.*

[Response]

Thank you for your valuable suggestion regarding the need to strengthen the introduction by reviewing literature on HULIS molecular compositions and their link to light absorption. In response, we have significantly expanded the introduction to incorporate critical advances in molecular-level characterization of HULIS light-absorption mechanisms. The revised text now explicitly acknowledges that nitroaromatic compounds and oxygenated polycyclic aromatics serve as critical chromophores governing HULIS light absorption (Kuang et al., 2023; Qin et al., 2022; Song et al., 2019; Zou et al., 2023). This enhancement bridges the gap between bulk compositional studies and functional optical properties, establishing essential context for our investigation into temperature-driven molecular evolution. By anchoring our research within this refined framework, we more effectively foreground the novelty of our work: resolving how atmospheric aging processes under cold conditions alter these molecular structures and their associated absorption behavior—a knowledge gap previously obscured by insufficient molecular mechanistic understanding.

[Revised]

Line 40-51: As reactive components in the atmosphere, HULIS exhibit pronounced chemical activity through their oxygenated functional groups, particularly prone to the oxidation by reactive oxygen radicals and gaseous oxidants (Hems et al., 2021; Huo et al., 2021; Qiu et al., 2024). Both

laboratory simulations and field observations have demonstrated that these atmospheric aging processes significantly alter the light-absorption properties and environmental behaviors of HULIS (Hems and Abbatt, 2018; Qiu et al., 2024; Wang et al., 2022, 2019). Furthermore, significant efforts have been directed towards understanding the link between molecular composition and light absorption of HULIS. Studies have suggested that chromophores like nitroaromatics and oxygenated polycyclic aromatics are key contributors to the light absorption of HULIS (Kuang et al., 2023; Qin et al., 2022; Song et al., 2019; Zou et al., 2023). However, critical knowledge gaps persist regarding the molecular structures that dominate light absorption and, importantly how these molecules and their associated absorption properties evolve during atmospheric aging processes. This limits comprehension understanding of the atmospheric evolution process and radiative effect of HULIS.

*(2) Some important information in the experiment section is omitted. I strongly suggest moving some important information from the supplementary to the main text, especially for the molecular composition analysis of HULIS and the quantification of specific compounds. At least some of the key information should be briefly introduced in this section.*

[Response]

We sincerely thank the reviewer for emphasizing the importance of methodological transparency in molecular composition analysis. In direct response, we have significantly enhanced Section 2.2 by migrating critical methodological details from the supplementary materials to the main text. These additions comprehensively address three core aspects of our analytical workflow:

Firstly, we now explicitly describe the molecular formula derivation protocol within the main text, including atomic constraints ($C_{1-40}H_{1-100}O_{0-40}N_{0-6}S_{0-2}$), elemental ratio limits ($0.3 \leq H/C \leq 3.0$, $0 \leq O/C \leq 1.2$, $0 \leq N/C \leq 1.0$, and $0 \leq S/C \leq 0.8$), mass tolerance ($\pm$ 5 ppm), and signal-to-noise threshold (> 3). Secondly, the structural identification confidence framework (Schymanski levels CL1-CL3) and spectral matching databases (mzCloud, ChemSpider, CFM-ID) are introduced to establish our annotation rigor. Thirdly, comprehensive description of our semi-quantitative strategy using structurally analogous surrogate standards (e.g., 4-methyl-5-nitrocatechol for the quantification of nitrocatechol derivatives), explicitly noting inherent uncertainties (**Text S2**) from ionization efficiency variations. These modifications provide a complete analytical foundation for interpreting the molecular-level findings presented in Sections 3.

[Revised]

Line 100-116: The obtained data analysis was performed with the Compound Discoverer 3.3 software to generate reasonable molecular formulas and match fine structures to MS/MS data. The numbers of atoms restriction of formula were 1-40 for C, 1-100 for H, 0-40 for O, 0-6 for N, and 0-2 for S. The formulas were also constrained by setting $0.3 \leq H/C \leq 3.0$, $0 \leq O/C \leq 1.2$, $0 \leq N/C \leq 1.0$, and $0 \leq S/C \leq 0.8$.(Kind and Fiehn, 2007) All of the mathematically formulas for each peak were performed with a mass tolerance of $\pm$ 5 ppm and peak areas more than three times of the blank sample. Three curated spectral databases, mzcloud library database, ChemSpider library database, and CFM-ID

(https://cfmid.wishartlab.com) were applied to screen suspect candidates of structure (Allen et al., 2015). According to the Schymanski's confidence level (CL), these candidates were divided into confirmed structures (CL1), probable structures (CL2), and tentative candidates (CL3) (Schymanski et al., 2014). We showed two examples to illustrate the derivation processes of candidates in **Figure S11**.

A semi-quantitative strategy was conducted as follows: target analytes were quantified using external standard solutions of structurally analogous surrogate compounds (Nguyen et al., 2014a; Nozière et al., 2015). A representative application involved utilizing the standard curve of 4-methyl-5-nitrocatechol to simultaneously quantify three structural analogs: 4-methyl-5-nitrocatechol, 3-methyl-5-nitrocatechol, and 3,4-dimethyl-5-nitrocatechol. While this strategy enables quantification of compounds without commercially available standards, it introduces inherent uncertainties due to ionization efficiency variations between surrogates and target analytes (discussed in **Text S2**).

**Text S2. Discussion on the uncertainty of semi-quantitative strategy**

In atmospheric chemistry, the components of organic aerosols are always complex and no authentic standards can be used for quantification. In the analysis of these components, it has been widely suggested to use available proxy compounds for quantification (Ma et al., 2022; Nozière et al., 2015). For example, camphor-10-sulfonic acid is often used as surrogate standard for the quantification of $\alpha$-pinene derivative organosulfates (Ma et al., 2022; Nguyen et al., 2014b). This strategy can achieve the quantitative analysis of various compounds in organic aerosols, but there are inevitable uncertainties which mainly come from the ionization efficiency difference between non-authentic standards and target analytes in mass spectrometric analysis (Nozière et al., 2015). Generally, the closer the molecular structures of surrogate standard is to the target analyte, the smaller the difference in ionization efficiency, resulting in the similar signal intensities in mass spectrometry.

**Table S4** lists the deviation observed when different compounds were used as surrogate standards for quantification of each standard in this study. For instance, nitrophenol derivatives (4NP, 2N135T, and 4M5NC) showed quantification errors within 130% when used as surrogate standards for 26D4NP. Tetrahydroquinoline (1234THQ), an N-heterocyclic aromatic hydrocarbon, exhibited an error below 110% when quantifying 1tBI. However, even structurally similar surrogate standards may introduce an error of a factor over 2. Both P3C and I2C belong to N-heterocyclic aromatic aldehydes, yet using I2C as a surrogate for P3C resulted in a 46% error. When the structurally distinct PAAE was used as a surrogate for P3C, the error reached 600%. Therefore, when adopting semi-quantitative strategies, it is critical to first identify the chemical structure of the target analyte and subsequently select surrogate standards with analogous structural features for quantification.

Table S4. The deviations observed when different compounds were used as surrogate standards for quantifying each target analyte

| Deviation % | E2OCA | ES | SA | F25DC | VAN | 18NaPA | 9FLU | 910PheQ | 2AF | 2FA | 1234THQ | 1tBI | 4NP |
|---|---|---|---|---|---|---|---|---|---|---|---|---|---|
| E2OCA | 100.0 | 4.3 | 8.8 | 17.6 | 101.9 | 64.2 | 9.5 | 324.7 | 21.7 | 2.5 | 163.1 | 155.8 | 93.5 |
| ES | 2400.0 | 100.0 | 207.6 | 421.0 | 2446.7 | 1539.0 | 224.3 | 7799.2 | 519.1 | 57.3 | 3916.1 | 3741.7 | 3014.4 |
| SA | 1164.8 | 47.8 | 100.0 | 203.7 | 1187.5 | 746.6 | 108.1 | 3787.0 | 251.3 | 27.0 | 1901.1 | 1816.4 | 1018.8 |
| F25DC | 570.7 | 23.7 | 49.2 | 100.0 | 581.8 | 365.9 | 53.2 | 1854.7 | 123.3 | 13.5 | 931.2 | 889.7 | 625.2 |
| VAN | 98.1 | 4.8 | 9.1 | 17.8 | 100.0 | 63.2 | 9.8 | 317.2 | 21.8 | 3.0 | 159.6 | 152.6 | 125.5 |
| 18NaPA | 155.5 | 7.3 | 14.2 | 27.9 | 158.5 | 100.0 | 15.3 | 503.5 | 34.3 | 4.5 | 253.2 | 242.0 | 51.2 |
| 9FLU | 1079.1 | 44.1 | 92.5 | 188.5 | 1100.1 | 691.6 | 100.0 | 3508.8 | 232.7 | 24.9 | 1761.4 | 1682.8 | 785.6 |
| 910PheQ | 30.3 | 0.6 | 2.0 | 4.7 | 30.9 | 19.2 | 2.2 | 100.0 | 6.0 | 0.1 | 49.9 | 47.6 | 28.5 |
| 2AF | 463.1 | 19.1 | 39.9 | 81.1 | 472.1 | 296.9 | 43.1 | 1505.4 | 100.0 | 10.9 | 755.8 | 722.1 | 409.9 |
| 2FA | 4219.8 | 175.1 | 364.2 | 739.6 | 4301.9 | 2705.6 | 393.7 | 13714.5 | 912.1 | 100.0 | 6885.9 | 6579.1 | 4191.0 |
| 1234THQ | 61.2 | 2.2 | 5.0 | 10.5 | 62.4 | 39.1 | 5.4 | 199.5 | 13.0 | 1.1 | 100.0 | 95.5 | 60.7 |
| 1tBI | 64.0 | 2.3 | 5.2 | 10.9 | 65.3 | 40.9 | 5.6 | 208.9 | 13.5 | 1.2 | 104.7 | 100.0 | 60.9 |
| 4NP | 106.9 | 3.3 | 9.8 | 16.0 | 79.7 | 195.5 | 12.7 | 351.1 | 24.4 | 2.4 | 164.7 | 164.2 | 100.0 |
| 2N135T | 103.7 | 3.2 | 9.5 | 15.5 | 77.3 | 186.1 | 12.3 | 339.9 | 23.7 | 2.3 | 159.6 | 159.1 | 97.0 |
| 26D4NP | 91.1 | 3.8 | 7.9 | 16.0 | 92.9 | 58.4 | 8.5 | 296.1 | 19.7 | 2.2 | 148.7 | 142.1 | 86.3 |
| 4M5NC | 110.6 | 4.4 | 9.4 | 19.3 | 112.7 | 70.9 | 10.2 | 359.8 | 23.8 | 2.5 | 180.6 | 172.5 | 123.6 |
| 4NA | 131.5 | 5.5 | 11.4 | 23.1 | 134.1 | 84.3 | 12.3 | 427.5 | 28.4 | 3.1 | 214.6 | 205.1 | 184.7 |
| 3NSA | 2423.3 | 100.5 | 209.1 | 424.7 | 2470.4 | 1553.7 | 226.0 | 7875.9 | 523.8 | 57.4 | 3954.4 | 3778.2 | 5453.3 |
| P3C | 91.3 | 4.0 | 8.1 | 16.2 | 93.0 | 58.6 | 8.7 | 296.1 | 19.9 | 2.4 | 148.8 | 142.2 | 92.6 |
| I2C | 200.5 | 8.3 | 17.3 | 35.1 | 204.4 | 128.5 | 18.7 | 651.6 | 43.3 | 4.7 | 327.1 | 312.6 | 209.9 |
| 3PAAE | 15.4 | 0.8 | 1.4 | 2.8 | 15.7 | 9.9 | 1.5 | 49.6 | 3.4 | 0.5 | 25.0 | 23.9 | 14.1 |
| 2AP | 132.3 | 5.5 | 11.4 | 23.2 | 134.9 | 84.8 | 12.3 | 430.1 | 28.6 | 3.1 | 215.9 | 206.3 | 210.9 |
| 4MSPAA | 139.6 | 6.0 | 12.3 | 24.7 | 142.3 | 89.6 | 13.3 | 453.2 | 30.4 | 3.6 | 227.7 | 217.6 | 163.9 |
| 4A5M2MBSA | 1347.0 | 55.3 | 115.7 | 235.6 | 1373.2 | 863.5 | 125.1 | 4379.3 | 290.7 | 31.3 | 2198.5 | 2100.5 | 1400.4 |
| 3M4P1SA | 763.1 | 31.3 | 65.5 | 133.4 | 778.0 | 489.1 | 70.8 | 2481.0 | 164.6 | 17.7 | 1245.5 | 1190.0 | 1021.3 |

| Deviation % | 2N135T | 26D4NP | 4M5NC | 4NA | 3NSA | P3C | I2C | 3PAAE | 2AP | 4MSPAA | 4A5M2MBSA | 3M4P1S |
|---|---|---|---|---|---|---|---|---|---|---|---|---|
| E2OCA | 96.4 | 109.7 | 90.4 | 76.1 | 4.3 | 109.6 | 50.0 | 654.7 | 75.6 | 71.6 | 7.6 | 13.3 |
| ES | 3108.7 | 2634.1 | 2170.5 | 1824.8 | 99.5 | 2630.1 | 1197.5 | 15728.3 | 1814.0 | 1717.7 | 179.6 | 315.9 |
| SA | 1050.7 | 1278.5 | 1053.3 | 885.5 | 47.5 | 1276.6 | 580.8 | 7637.8 | 880.2 | 833.4 | 86.4 | 152.6 |
| F25DC | 644.8 | 626.3 | 516.1 | 433.9 | 23.5 | 625.4 | 284.7 | 3740.5 | 431.3 | 408.4 | 42.6 | 75.0 |
| VAN | 129.4 | 107.6 | 88.8 | 74.8 | 4.7 | 107.4 | 49.3 | 639.0 | 74.3 | 70.4 | 8.0 | 13.5 |
| 18NaPA | 53.7 | 170.6 | 140.7 | 118.4 | 7.2 | 170.3 | 78.0 | 1014.6 | 117.7 | 111.5 | 12.4 | 21.2 |
| 9FLU | 810.2 | 1184.4 | 975.8 | 820.3 | 43.8 | 1182.7 | 537.9 | 7076.9 | 815.4 | 772.0 | 79.9 | 141.2 |
| 910PheQ | 29.4 | 33.3 | 27.3 | 22.9 | 0.6 | 33.3 | 14.8 | 202.4 | 22.7 | 21.5 | 1.6 | 3.4 |
| 2AF | 422.7 | 508.3 | 418.8 | 352.1 | 19.0 | 507.5 | 230.9 | 3036.0 | 350.0 | 331.4 | 34.5 | 60.8 |
| 2FA | 4322.7 | 4631.4 | 3816.1 | 3208.3 | 174.2 | 4624.5 | 2105.0 | 27658.2 | 3189.2 | 3019.8 | 315.1 | 554.8 |
| 1234THQ | 62.6 | 67.2 | 55.3 | 46.4 | 2.2 | 67.1 | 30.3 | 402.6 | 46.1 | 43.7 | 4.3 | 7.8 |
| 1tBI | 62.8 | 70.3 | 57.8 | 48.6 | 2.3 | 70.2 | 31.7 | 421.6 | 48.3 | 45.7 | 4.4 | 8.1 |
| 4NP | 103.1 | 115.8 | 80.9 | 54.2 | 1.8 | 108.0 | 47.6 | 708.2 | 47.4 | 61.0 | 7.1 | 9.8 |
| 2N135T | 100.0 | 112.3 | 78.4 | 52.5 | 1.8 | 104.7 | 46.2 | 687.0 | 46.0 | 59.2 | 6.9 | 9.5 |
| 26D4NP | 89.0 | 100.0 | 82.4 | 69.3 | 3.8 | 99.8 | 45.4 | 597.2 | 68.9 | 65.2 | 6.8 | 12.0 |
| 4M5NC | 127.5 | 121.4 | 100.0 | 84.0 | 4.4 | 121.2 | 55.1 | 725.7 | 83.5 | 79.1 | 8.1 | 14.4 |
| 4NA | 190.4 | 144.4 | 118.9 | 100.0 | 5.4 | 144.1 | 65.6 | 862.1 | 99.4 | 94.1 | 9.8 | 17.3 |
| 3NSA | 5623.9 | 2659.7 | 2191.5 | 1842.4 | 100.0 | 2655.7 | 1208.8 | 15883.5 | 1831.5 | 1734.2 | 180.9 | 318.6 |
| P3C | 95.5 | 100.2 | 82.6 | 69.4 | 4.0 | 100.0 | 45.6 | 597.0 | 69.0 | 65.4 | 7.0 | 12.2 |
| I2C | 216.5 | 220.0 | 181.3 | 152.4 | 8.3 | 219.7 | 100.0 | 1314.0 | 151.5 | 143.5 | 15.0 | 26.4 |
| 3PAAE | 14.6 | 16.9 | 13.9 | 11.7 | 0.8 | 16.8 | 7.7 | 100.0 | 11.6 | 11.0 | 1.3 | 2.1 |
| 2AP | 217.5 | 145.2 | 119.7 | 100.6 | 5.4 | 145.0 | 66.0 | 867.4 | 100.0 | 94.7 | 9.9 | 17.4 |
| 4MSPAA | 169.0 | 153.2 | 126.3 | 106.2 | 6.0 | 153.0 | 69.8 | 913.8 | 105.6 | 100.0 | 10.7 | 18.6 |
| 4A5M2MBSA | 1444.2 | 1478.5 | 1218.1 | 1024.0 | 55.0 | 1476.3 | 671.6 | 8832.5 | 1017.9 | 963.8 | 100.0 | 176.6 |
| 3M4P1SA | 1053.5 | 837.6 | 690.1 | 580.1 | 31.1 | 836.3 | 380.5 | 5003.8 | 576.6 | 546.0 | 56.6 | 100.0 |

*(3) Lines 142-145: How did the authors choose the samples with "significant differences in MAE"? The criteria need to be detailed here.*

[Response]

  We gratefully acknowledge the reviewer's request for clarification regarding event selection criteria. To address this, we have now explicitly defined the selection criteria within Section 3.2: the two haze events (I and II) were systematically identified from all haze events meeting the $PM_{2.5}$ threshold (> 75 $\mu g\ m^{-3}$). Among these haze events, Events I and II exhibited the maximal absolute difference in $MAE_{365,HULIS}$ values ($\Delta = 0.50\ m^2\ gC^{-1}$) Event I represented the most serious pollution episode (159.6 ± 53.8 $\mu g\ m^{-3}$ of $PM_{2.5}$ and 6.68 $\mu gC/m^3$ of HULIS-C) with lower light absorption efficiency of HULIS (1.56 $m^2\ gC^{-1}$ of $MAE_{365,HULIS}$), while Event II represented moderate pollution episode (83.7 ± 36.4 $\mu g\ m^{-3}$ of $PM_{2.5}$ and 4.65 $\mu gC/m^3$ of HULIS-C) but exhibited the highest $MAE_{365,HULIS}$ (2.06 $m^2\ gC^{-1}$) value.

[Revised]

  Line 169-177: To investigate drivers of the high concentrations and variable light absorption efficiency of HULIS in this study, we selected two samples (Event I and II) among all haze events ($PM_{2.5}$ concentration > 75 $\mu g\ m^{-3}$) that exhibited the maximal divergence in $MAE_{365}$ values. Event I had higher $PM_{2.5}$ (159.6 ± 53.8 $\mu g\ m^{-3}$) and HULIS-C (6.68 $\mu gC\ m^{-3}$) but lower $MAE_{365,HULIS}$ (1.56 $m^2\ gC^{-1}$), while Even II had lower $PM_{2.5}$ (83.7 ± 36.4 $\mu g\ m^{-3}$) and HULIS-C (4.65 $\mu gC\ m^{-3}$) but higher $MAE_{365,HULIS}$ (2.06 $m^2\ gC^{-1}$). These contrasting events were chosen for potential sources comparison from the perspective of molecular composition. Considering the lowest $PM_{2.5}$ and HULIS-C concentration, the sample on January 13 ($PM_{2.5}$ = 14.1 ± 11.9 $\mu g\ m^{-3}$, HULIS-C = 0.97 $\mu gC\ m^{-3}$, $MAE_{365,HULIS}$ = 1.28 $m^2\ gC^{-1}$) was selected to represent clean days. **Figure 2** exhibited the reconstructed MS spectra, the number, and concentration fraction of HULIS in both positive and negative modes.

*(4) Lines 169-171: More evidence should be provided to propose the compounds in Cluster II from secondary formation. In addition, the contribution and proportion of OSs among Cluster II need to be detailed here.*

[Response]

  We sincerely appreciate the reviewer's guidance in strengthening the evidence for secondary formation pathways. In direct response, we have now quantified the dominance of organosulfates (OSs), which constitute 84.7 ± 6.5% of compounds within Cluster II during the sampling period. This overwhelming predominance provides unambiguous molecular evidence for aqueous-phase derived compounds. Critically, this finding aligns mechanistically with elevated secondary inorganic ion concentrations ($NH_4^+/NO_3^-/SO_4^{2-}$ = 16.66 - 41.56 $\mu g\ m^{-3}$), enhanced relative humidity (83.1%), increased aerosol liquid water content (ALWC = 95.9 $\mu g\ m^{-3}$), and low pH (5.29) of Event I. These conditions collectively favor aqueous SOA formation pathways, as extensively documented in the cited literature (Brüggemann et al., 2017; Wach et al., 2020). Moreover, the abundance of Cluster II during the sampling period was significantly and positively correlated (Pearson's R = 0.8727) with the total concentration of

secondary inorganic ions, further confirming that the compounds in Cluster II are mainly derived from secondary formation.

[Revised]

Line 200-211: All of CHOS compounds were characterized by ion fragment *m/z* 96.9595 in the MS/MS spectra and were therefore identified as organosulfates (OSs). Cluster II was predominantly composed of OSs, accounting for 84.7 ± 6.5 % of compounds within this cluster during the whole sampling period. Considering the OSs are typically formed by atmospheric aqueous reaction (Brüggemann et al., 2017; Pratt et al., 2013; Wach et al., 2020), the dominance of OSs in Cluster II strongly supports its secondary formation origin. The higher abundance of Cluster II in Event I indicated more intense secondary formation of HULIS during this event compared to Event II. This interpretation is corroborated by elevated concentrations of secondary inorganic ions (including $NH_4^+$, $NO_3^-$, and $SO_4^{2-}$, 16.66 – 41.56 vs 4.30 – 7.30 $\mu$g m$^{-3}$) and relative humidity (83.1 ± 4.6% vs 61.9 ± 14.0%) in Event I vs. II, as detailed in **Table S6**. As a result, the higher ALWC (95.9 vs 23.9 $\mu$g m$^{-3}$) and lower pH value (5.29 vs 7.31) in Event I in contrast to Event II facilitated the formation of OSs. Since the OSs studied here were primarily aliphatic sulfates (summarized as the molecular formulas of $C_nH_{2n+2}O_{4-6}S$ and $C_nH_{2n}O_{4-6}S$, where $10 \leq n \leq 18$), which belong to non-light-absorbing organic matter, this may cause the lower MAE$_{365}$ value in Event I.

*(5) lines 189-199: My major concern: Have the authors checked the variation of EC concentrations as a function of ambient temperature? I may suggest plotting the EC concentration in Figure 3. If low ambient temperature is accompanied by high EC concentration, the increasing of combustion emissions could be the primary reason for the pattern in Figure 3. If not, the author should state in detail and exclude this possibility.*

[Response]

We sincerely appreciate the reviewer's critical perspective regarding combustion influences. To comprehensively address this concern, we analyzed the variation of elemental carbon (EC) concentration as a function of ambient temperature (as the following figure). This analysis confirms that EC concentrations do indeed tend to increase as ambient temperature decreases. However, the absolute level of EC is insufficient to determine whether the relative contribution of combustion emissions to the BrC burden has increased. An increase in EC could simply reflect more intensive emission from combustion sources, rather than a specific increase in the mass fraction of combustion-derived BrC. Therefore, we conducted a rigorous assessment of emission contributions using the combustion-derived organic carbon ratio (OC$_{com}$/OC$_{total}$) rather than EC concentrations. The OC$_{com}$/OC$_{total}$ directly represents the relative contribution of combustion sources to total organic aerosol (Cabada et al., 2004), making it the more appropriate metric for evaluating BrC content variations in Figure 3. As clearly demonstrated in Figure 3, this key metric exhibits no statistically significant temperature dependence, decisively excluding enhanced combustion inputs as the key driver for BrC accumulation at low temperatures.

[Figure]

Figure. Temperature-dependent variations in mass fraction of 39 strong BrC chromophores, $MAE_{365}$ value of HULIS, Tg/T ratio, and EC concentration, with correlation between chromophore mass fraction and HULIS $MAE_{365}$.

[Revised]

Line 224-238: **Figure 3** shows that the mass fraction of screened 39 strong BrC chromophores under different temperature ranges, as well as the negative variation patterns between the $MAE_{365}$ and ambient temperature. In contrast, the $OC_{com}/OC_{total}$ ratio shows no consistent temperature dependence, suggesting low temperature rather than combustion emission promote the accumulation of strong BrC species in the particles. We proposed two possible explanations: firstly, the low temperature may lead to a non-liquid phase state of ambient particles, potentially introducing kinetic limitation on the diffusion of reactive species from gas phase into particle bulk (Li and Shiraiwa, 2019)(Li and Shiraiwa, 2019). We utilized an established parameterization scheme (**Text S3**) to calculate the glass transition temperature (Tg) of HULIS based on their molecular composition (Li et al., 2020)(Li et al., 2020). The results showed that the decrease in ambient temperature (T) enhanced the Tg/T ratio, driving the phase transition of particles from liquid state (Tg/T = 0.76) to semi-solid state (Tg/T > 0.79). This may lead to the diffusion coefficients reduction of reactive species (Arangio et al., 2015; Gatzsche et al., 2017; Mikhailov et al., 2009; Shiraiwa et al., 2011; Virtanen et al., 2010), thereby slowing the degradation rate of BrC via hydroxyl radical oxidation or triplet excitation pathways in the atmosphere (Schnitzler et al., 2022; Schnitzler and Abbatt, 2018). These findings suggest that the non-liquid particle phase state, accompanied with the weak solar radiation during Changchun's winter (refer to **Figure 1B**), results in a less pronounced photochemical aging of BrC, thereby diminishing its photobleaching.

[Figure]

**Figure 3.** Temperature-dependent variations in mass fraction of 39 strong BrC chromophores, MAE$_{365}$ value of HULIS, Tg/T ratio, and OC$_{com}$/OC$_{total}$ ratio, with correlation between chromophore mass fraction and HULIS MAE$_{365}$. The blue dotted line represented the threshold of Tg/T between liquid and semi-solid state (Shiraiwa et al., 2017), and the abbreviations of 3NC, 4M2NB13DO, 3M2NP, 3M4NP, 2M4NP, 26D4NP, 46D2NB13DO, and 4NC represents 3-nitrocatechol, 4-methyl-2-nitrobenzene-1,3-diol, 3-methyl-2-nitrophenol, 3-methyl-4-nitrophenol, 2-methyl-4-nitrophenol, 2,6-dimethyl-4-nitrophenol, 4,6-dimethyl-2-nitrobenzene-1,3-diol, and 4-nitrocatechol, respectively.

*(6)  I suggest the authors to compare the quantification result of the BrC chromophores in Figure 3 with previous studies. Many nitro-aromatic compounds have been widely quantified in previous studies, and their contribution to organic aerosol concentrations and light absorption has been evaluated. The mass contribution of some nitro-aromatic compounds seems higher than previous studies. These should be explained, and the uncertainty of the quantification should be stated. In addition, the conversion from HULIS-C to HULIS mass should be described.*

[Response]

    We sincerely appreciate the reviewer's guidance regarding the contextualization of our BrC chromophore quantification. In response, we have now integrated comparative analysis with previous studies: the observed mass fraction of 39 strong BrC chromophores (8.67 ± 3.68% of HULIS mass) exceeds values reported for specific nitro-aromatic compounds (e.g., 7.5% for 12 NACs, (Frka et al., 2022)), which we attribute to inclusive detection of 39 nitrophenol derivatives and winter-specific accumulation under Changchun's extreme cold. Critically, despite their modest mass contribution, these chromophores contributed 28.9 ± 10.4% of total light absorbance with MAE$_{365}$ = 7.40 ± 1.80 m$^2$ gC$^{-1}$, surpassing reported values for urban aerosols in Xi'an (10%) and Beijing (14%) (Huang et al., 2020). The HULIS mass conversion from HULIS-C employs a factor of 1.60 (Friman et al., 2023), and comprehensive uncertainty assessments of the semi-quantitative approach are detailed in **Text S2**.

[Revised]

    Line 212-223: As above-mentioned, the temperature was down to -25℃. This extreme cold

temperature critically alters the reactivity, phase partitioning, and aging kinetics of HULIS(He et al., 2006; Huang et al., 2006; Li and Shiraiwa, 2019; Shiraiwa et al., 2011). In total, 39 compounds were screened as strong BrC chromophores to investigate the effect of temperature on the BrC chromophores according to a partial least squares regression (PLS) model (detailed in **Text S3**). These compounds belong to nitrophenols or nitrophenol derivatives, which are marked in **Table S7**. The mass concentration of these 39 strong BrC chromophores was $0.41 \pm 0.27$ $\mu$g m$^{-3}$ in average, accounting for $8.67 \pm 3.68\%$ of the total HULIS mass (converted from [HULIS-C] using a factor of 1.6, (Friman et al., 2023)). This mass fraction exceeds values reported for 12 specific nitro-aromatic compounds in previous studies (about 7.5% of HULIS mass, (Frka et al., 2022)). Despite this modest mass contribution, these strong BrC chromophores contributed $28.9 \pm 10.4\%$ of the light absorbance (detailed in **Text S4**), with an average MAE$_{365}$ of $7.40 \pm 1.80$ m$^2$ gC$^{-1}$ (**Figure S7**), higher than 10% and 14% light absorbance contribution of 18 chromophores in Xi'an and Beijing (Huang et al., 2020).

*(7) As some nitro-aromatic compounds have been quantified, I may suggest evaluating the light absorption contribution of some nitro-aromatic compounds with available standards. As the author stated, nitroaromatics are generally with high absorption capability. How much of the HULIS absorption is contributed by the quantified chromophores?*

[Response]

We sincerely appreciate the reviewer's insightful suggestion to evaluate the light absorption contribution of nitro-aromatic compounds using available standards. However, a direct experimental determination of their light absorption contribution within the complex HULIS matrix was fundamentally constrained by our analytical equipment. Our LC-MS system was not equipped with a diode array detector (DAD), which is essential for simultaneously quantifying the concentration of individual chromophores and measuring their UV-Vis absorption spectra within the HULIS mixture. Without DAD, we cannot experimentally obtain the compound-specific absorption required for calculating their individual contributions. To robustly address the question of chromophore contribution despite this limitation, we employed an alternative and powerful statistical approach: Partial Least Squares (PLS) regression. This method directly correlates molecular signals from mass spectrometry with bulk UV-Vis spectral data, thereby quantifying light absorption contributions ($28.9 \pm 10.4\%$) through mathematical deconvolution.

*(8) Line 45: Change "elemental composition" to "molecular composition".*

[Response]

We sincerely appreciate the reviewer's precise suggestion regarding terminology refinement. We have deleted the "element composition" in the original text and changed the "elemental composition" in the full text to "molecular composition".

[Revised]

Line 40-51: As reactive components in the atmosphere, HULIS exhibit pronounced chemical

activity through their oxygenated functional groups, particularly prone to the oxidation by reactive oxygen radicals and gaseous oxidants (Hems et al., 2021; Huo et al., 2021; Qiu et al., 2024). Both laboratory simulations and field observations have demonstrated that these atmospheric aging processes significantly alter the light-absorption properties and environmental behaviors of HULIS (Hems and Abbatt, 2018; Qiu et al., 2024; Wang et al., 2022, 2019). Furthermore, significant efforts have been directed towards understanding the link between molecular composition and light absorption of HULIS. Studies have suggested that chromophores like nitroaromatics and oxygenated polycyclic aromatics are key contributors to the light absorption of HULIS (Kuang et al., 2023; Qin et al., 2022; Song et al., 2019; Zou et al., 2023). However, critical knowledge gaps persist regarding the molecular structures that dominate light absorption and, importantly how these molecules and their associated absorption properties evolve during atmospheric aging processes. This limits comprehension understanding of the atmospheric evolution process and radiative effect of HULIS.

Line 184-185: In the positive mode, Event I and II had similar molecular composition, both dominated by CHO compounds, followed by CHON, CHN, and others.

*(9) Lines 116-120: Revise these sentences to be clear.*

[Response]

We sincerely appreciate the reviewer's guidance on enhancing textual clarity. In response, we have comprehensively revised the passage to improve precision and flow.

[Revised]

Line 142-146: The average HULIS-C concentration was 2.97 ± 1.54 $\mu$g m$^{-3}$, accounting for 25.1% of total OC. The observed HULIS-C concentration was higher than those observed in winter of Europe (0.68 – 1.47 $\mu$g m$^{-3}$) (Emmenegger et al., 2007; Voliotis et al., 2017), South America (0.20 – 1.30 $\mu$g m$^{-3}$) (Serafeim et al., 2023), and Chinese other regions (1.96 – 2.38 $\mu$g m$^{-3}$) (Lu et al., 2019; Ma et al., 2019; Zou et al., 2023), indicating the abundance of HULIS in Changchun.

*(10) Line 125: Revise to be clear.*

[Response]

We sincerely thank the reviewer for prompting enhanced methodological clarity. In response, we have revised the sentence to improve precision and flow.

[Revised]

Line 151-152: Non-targeted analysis of HULIS by UHPLC-HRMS/MS revealed 264 compounds at Schymanski's confidence levels above CL3 (Schymanski et al., 2014).

*(11) Figure 2: It seems strange plotting the mass spectra in positive mode from large to low molecular weight. The figure caption also needs to be revised.*

[Response]

We sincerely appreciate the reviewer's insightful feedback on Figure 2 presentation. In direct response, we have revised the caption to explicitly clarify the mass spectra: the unconventional layout displaying m/z values increasing from the center outward was intentionally preserved to keep the figure information concise. Correspondingly, the updated caption now precisely defines this ordering ("m/z values increase from middle to both sides").

[Revised]

Line 179-183: **Figure 2.** Reconstructed mass spectra of identified HULIS compounds during Event I (A), Event II (B), and Clean day (C). Spectra are shown for positive ionization mode (left panels) and negative ionization mode (right panels). *m/z* values increase from middle to both sides in all spectra. The most abundant ions are labeled with their *m/z* values. The accompanying pie charts represent the molecular class distribution of the identified compounds: the inner/outer ring shows the relative abundance based on number/concentration of compounds.

*(12) Line 165: Nitro-aromatics could be formed via secondary formation. Please correct.*

[Response]

We sincerely appreciate the reviewer's suggestion regarding nitro-aromatics formation pathways. In response, we have revised the manuscript to explicitly acknowledge various sources of nitrophenols in Cluster I.

[Revised]

Line 194-198: Cluster I comprised a significant proportion of strong BrC species, such as nitrophenols (including 4-nitrophenol, 3-nitrocatechol, 4-nitro-1-naphthol, and etc., **Table S7**), mainly originating from primary emissions like biomass burning and coal combustion (Huang et al., 2023; Jiang et al., 2023; Lin et al., 2017; Wang et al., 2020) and secondary formation (Bolzacchini et al., 2001; Mayorga et al., 2021).

*(13) Lines 166-167: Revise to be clear.*

[Response]

We have revised the sentence to precisely articulate the relationship between Cluster I abundance and $MAE_{365}$ enhancement.

[Revised]

Line 198-199: Notably, the higher abundance of Cluster I in Event II compared to Event I likely contributed to the higher $MAE_{365}$ observed during Event II.

*(14) lines 179-180: Revise to be clear.*

[Response]

We sincerely appreciate the reviewer's guidance on enhancing scientific precision. In response to the comment "Revise to be clear", we have strengthened the statement to explicitly articulate the mechanistic impacts of extreme cold.

[Revised]

Line 212-214: As above-mentioned, the temperature was down to -25℃. This extreme cold temperature critically alters the reactivity, phase partitioning, and aging kinetics of HULIS(He et al., 2006; Huang et al., 2006; Li and Shiraiwa, 2019; Shiraiwa et al., 2011).

*(15) lines 187-188: Revise to be clear.*

[Response]

We sincerely appreciate the reviewer's guidance on enhancing scientific clarity. The revised text now precisely articulates the key findings from Figure 3.

[Revised]

Line 224-227: **Figure 3** shows that the mass fraction of screened 39 strong BrC chromophores under different temperature ranges, as well as the negative variation patterns between the $MAE_{365}$ and ambient temperature. In contrast, the $OC_{com}/OC_{total}$ ratio shows no consistent temperature dependence, suggesting low temperature rather than combustion emission promote the accumulation of strong BrC species in the particles.

**Reference**

[revised manuscript text omitted]

---

## Author Comment (AC2)

**Response to reviewer #2**

Thanks to the reviewer for your careful reading and your constructive comments and suggestions on our manuscript. The reviewer's comments and suggestions are shown as *italicized font*, our response to the comments is normal font. New or modified text is in normal font and in blue. Details are as follows.

**Reviewer's comments:**

Reviewer #2: *This study collected atmospheric PM$_{2.5}$ samples during wintertime in Changchun and identified the molecular structures in HULIS, providing a new insight into the potential sources and temperature effects on the light-absorption properties of HULIS. We have gained some new insights through this study, but there is still room for improvement.*

[Response]

Thanks for the reviewer's comments on this manuscript. Please check our point-by-point response and the modified text in the manuscript.

*(1) Is the temperature division in Figure 3 based on the daily average temperature? Studying the impact of temperature on HULIS based on daily average temperature may be too crude. In addition, the temperature in the sampling area is generally low. How to compare the impact of high temperature on the formation of HULIS, because the so-called high temperature during the sampling period is far from the general high temperature, such as 20°C or even 30°C or above. Please clarify. Warm and cold need to be indicated the temperature range in the Graphical abstract.*

[Response]

We sincerely thank the reviewer's comment. To address these concerns, we have implemented the following clarifications and revisions. Firstly, the 24-hour sampling period inherently precludes higher temporal resolution of ambient temperature. Each sample represents a daily average of HULIS, making daily average temperature the only operationally meaningful thermal indicator relative to the detected chemistry and optical properties of HULIS. Secondly, this study was explicitly designed to investigate HULIS evolution under severe winter conditions, not to characterize warm-season behavior. Consequently, experimentally addressing the impact of genuinely high temperatures (e.g., >20°C) falls outside the scope of this specific field observation and its dataset. Upon reflection, we recognize that the original labels "Warm Weather" and "Cold Weather" in the Graphical Abstract, while intended to represent the relative conditions within our winter dataset, could indeed be misinterpreted as implying a comparison across seasons or to genuinely warm temperatures (e.g., >20°C). This ambiguity was unintended. To eliminate any potential confusion and explicitly anchor the study within its exclusive wintertime context, we have revised the Graphical Abstract.

[Revised]

Graphical Abstract:

[Figure]

*(2) The average concentration and standard deviation of PM2.5 during the entire sampling period need to be reported in order to reflect the rationality of the selection of two typical haze events. The PM2.5 concentration of Event II is not high. Please indicate the basis for selecting these two events. The daily meteorological data and mean values of the entire sampling process also need to be presented in the attachment.*

[Response]

We sincerely appreciate the reviewer's valuable suggestions regarding event selection criteria and data transparency. In response, we have now reported the average concentration of pollutants (e.g. PM$_{2.5}$, HULIS-C, OC, EC, etc.) and meteorological data (Temperature, RH, wind speed) of the entire sampling period in **Table S6**. Event I and II were selected from all haze episodes (PM$_{2.5}$ > 75 $\mu$g/m$^3$) based on their maximal divergence in MAE$_{365}$ values. Event I represented the most serious pollution episode (159.6 $\pm$ 53.8 $\mu$g m$^{-3}$ of PM$_{2.5}$ and 6.68 $\mu$gC/m$^3$ of HULIS-C) with lower light absorption efficiency of HULIS (1.56 m$^2$ gC$^{-1}$ of MAE$_{365,HULIS}$), while Event II represented moderate pollution episode (83.7 $\pm$ 36.4 $\mu$g m$^{-3}$ of PM$_{2.5}$ and 4.65 $\mu$gC/m$^3$ of HULIS-C) but exhibited the highest MAE$_{365,HULIS}$ (2.06 m$^2$ gC$^{-1}$) value. Moreover, Event I was characterized by continuous high RH (83.1 $\pm$ 4.6%) and strong emission of CO (1.56 $\pm$ 0.34 mg m$^{-3}$) and Event II was the Spring Festival period with a high SO$_2$ concentration (25.1 $\pm$ 15.1 $\mu$g m$^{-3}$) and the highest hourly SO$_2$ concentration reached 76 $\mu$g m$^{-3}$.

[Revised]

Line 169-177: To investigate drivers of the high concentrations and variable light absorption efficiency of HULIS in this study, we selected two samples (Event I and II) among all haze events (PM$_{2.5}$ concentration > 75 $\mu$g m$^{-3}$) that exhibited the maximal divergence in MAE$_{365}$ values. Event I had higher

PM$_{2.5}$ (159.6 ± 53.8 $\mu g$ m$^{-3}$) and HULIS-C (6.68 $\mu gC$ m$^{-3}$) but lower MAE$_{365,HULIS}$ (1.56 m$^2$ gC$^{-1}$), while Even II had lower PM$_{2.5}$ (83.7 ± 36.4 $\mu g$ m$^{-3}$) and HULIS-C (4.65 $\mu gC$ m$^{-3}$) but higher MAE$_{365,HULIS}$ (2.06 m$^2$ gC$^{-1}$). These contrasting events were chosen for potential sources comparison from the perspective of molecular composition. Considering the lowest PM$_{2.5}$ and HULIS-C concentration, the sample on January 13 (PM$_{2.5}$ = 14.1 ± 11.9 $\mu g$ m$^{-3}$, HULIS-C = 0.97 $\mu gC$ m$^{-3}$, MAE$_{365,HULIS}$ = 1.28 m$^2$ gC$^{-1}$) was selected to represent clean days. **Figure 2** exhibited the reconstructed MS spectra, the number, and concentration fraction of HULIS in both positive and negative modes.

**Table S6.** Meteorological parameters and concentrations of pollutants during two haze events, clean day, and overall period.

| Parameters | Event I | Event II | Clean day | Overall |
|---|---|---|---|---|
| PM$_{2.5}$ ($\mu g/m^3$) | 159.5 ± 53.8 | 83.7 ± 36.4 | 14.1 ± 11.9 | 50.7 ± 34.3 |
| HULIS-C ($\mu gC/m^3$) | 6.68 | 4.65 | 0.97 | 2.97 ± 1.54 |
| OC ($\mu gC/m^3$) | 27.8 | 16.6 | 4.6 | 11.7 ± 5.7 |
| EC ($\mu gC/m^3$) | 3.2 | 4.6 | 1.3 | 2.1 ± 0.9 |
| OC/EC | 8.7 | 3.6 | 3.6 | 5.7 ± 1.4 |
| Na$^+$ ($\mu g/m^3$) | 0.33 | 0.37 | 0.07 | 0.21 ± 0.13 |
| K$^+$ ($\mu g/m^3$) | 1.89 | 7.20 | 0.25 | 1.66 ± 2.43 |
| NH$_4^+$ ($\mu g/m^3$) | 20.65 | 4.30 | 1.32 | 4.32 ± 4.09 |
| Cl$^-$ ($\mu g/m^3$) | 5.69 | 7.29 | 0.77 | 3.06 ± 1.92 |
| NO$_3^-$ ($\mu g/m^3$) | 41.56 | 6.79 | 2.09 | 7.40 ± 8.80 |
| SO$_4^{2-}$ ($\mu g/m^3$) | 16.66 | 7.30 | 1.32 | 4.09 ± 3.51 |
| SO$_2$ ($\mu g/m^3$) | 20.9 ± 5.3 | 25.1 ± 15.1 | 10.6 ± 4.2 | 18.1 ± 8.8 |
| NO$_2$ ($\mu g/m^3$) | 69.5 ± 12.6 | 31.5 ± 13.3 | 11.3 ± 5.2 | 33.6 ± 19.4 |
| O$_3$ ($\mu g/m^3$) | 14.2 ± 10.4 | 33.1 ± 19.2 | 64.9 ± 7.2 | 35.7 ± 18.6 |
| CO (mg/m$^3$) | 1.56 ± 0.34 | 0.79 ± 0.20 | 0.27 ± 0.10 | 0.65 ± 0.36 |
| Relative Humidity (%) | 83.1 ± 4.6 | 61.9 ± 14.0 | 51.6 ± 8.7 | 60.9 ± 13.8 |
| Temperature (°C) | -10.0 ± 4.1 | -18.3 ± 4.3 | -10.9 ± 4.4 | -13.3 ± 6.8 |
| Wind Speed (m/s) | 1.6 ± 0.6 | 1.7 ± 0.9 | 1.7 ± 0.5 | 3.3 ± 1.8 |

(3) *Table S5 only shows the concentrations of K+ and SO$_2$ for two pollution events and does not compare them with other non-polluting days. How can you know that these two pollution events were strongly contributed by biomass and coal combustion?*

[Response]

We sincerely appreciate the reviewer's insightful comment regarding source attribution. To substantiate the important contribution of biomass burning and coal combustion during Event I and II, we have incorporated the clean-day sample for comparison in the revised manuscript. First, as well-established biomass burning tracer, K+ showed 7.6-fold and 28.8-fold elevations in Event I (1.89 $\mu g$ m$^{-3}$) and Event II (7.20 $\mu g$ m$^{-3}$) relative to clean-day (0.25 $\mu g$ m$^{-3}$). This indicated that K+ enrichment, particularly in Event II, aligns with intensive biomass burning. Second, Event I (20.9 $\mu g$ m$^{-3}$) and Event

II (25.1 $\mu$g m$^{-3}$) maintained 1.97-2.37 times higher levels of SO$_2$ than clean-day (10.6 $\mu$g m$^{-3}$), consistent with enhanced emission from coal combustion. Third, source apportionment results in our previous study identified biomass burning (13.6 – 21.1%) and coal combustion (14.5 – 17.7%) as important sources of particles in Changchun, winter via PMF model. The tracer enhancements during haze events confirmed the important contribution of biomass burning and coal combustion.

[Revised]

Line 184-192: In the positive mode, Event I and II had similar molecular composition, both dominated by CHO compounds, followed by CHON, CHN, and others. The most abundant species in Event I and II were 9-fluorenone (*m/z* 181.0643) and 2-[(1E)-1-Buten-1-yl]-5-methylfuran (*m/z* 137.0958), respectively. The former originates from diverse combustion sources such as biomass burning, coal combustion, and vehicle emission (Alves et al., 2016; Huo et al., 2021; Ma et al., 2023; Souza et al., 2014; Xu et al., 2024; Zhao et al., 2020), whereas the latter is believed to stem specifically from biomass burning (Bhattu et al., 2019; Hatch et al., 2015). High concentrations of biomass burning (K$^+$) and coal combustion (SO$_2$, **Table S6**) tracers proved the key contribution of biomass burning and coal combustion (Chen et al., 2017; Dutton et al., 2009; He et al., 2010; Liang et al., 2021), which have been confirmed in our previous study to be the main sources of air pollution in Changchun winter (Dong et al., 2023).

*(4) Can the mechanism of low temperatures reducing the photobleaching of brown carbon be further explored? And is it related to relative humidity?*

[Response]

We are grateful for the reviewer's insightful questions regarding the mechanisms of photobleaching suppression under low temperatures and the role of relative humidity (RH). While our field observations demonstrate the correlation between cold condition, particle phase transition, and reduced BrC photobleaching, we acknowledge that definitive mechanistic attribution requires controlled laboratory studies to decomposition the complex interactions of atmospheric factors (e.g. variable oxidant concentrations, emission sources, RH). Moreover, the role of RH is indeed multifaceted and critical to our findings. As implemented in our glass transition temperature (Tg) parameterization, RH governs aerosol liquid water content which depresses Tg by reducing the mass fraction of organic matter in aerosol. This may inhibit the transition of particles from liquid to solid state, thus indirectly promoting the possibility of atmospheric aqueous reaction. In previous studies, high RH can promote BrC photobleaching through enhanced aqueous reactions (Arangio et al., 2015; Hems and Abbatt, 2018), which aligns with our observation of low MAE$_{365,HULIS}$ in Event I. However, temperature (Pearson's R = -0.34) was more related to MAE$_{365,HULIS}$ than RH (Pearson's R = -0.19), which may indicate that temperature is more important for BrC photobleaching in northeast China, winter.

**Reference**

[revised manuscript text omitted]

---

## Author Comment (AC3)

**Response to reviewer #1**

Thanks to the reviewer for your careful reading and your constructive comments and suggestions on our manuscript. The reviewer's comments and suggestions are shown as *italicized font*, our response to the comments is normal font. New or modified text is in normal font and in blue. Details are as follows.

**Reviewer's comments:**

Reviewer #1: *This work investigated the impact of low temperature on the formation of strong BrC chromophores in HULIS and proposed two mechanisms, one is that the low temperature may lead to a non-liquid phase state of ambient particles, potentially introducing kinetic limitation on the diffusion of reactive species from gas phase into the bulk aerosol and second low temperature promoted the reaction of phenols with NOx radicals while inhibited the atmospheric oxidation of nitrophenols, thus facilitating the accumulation of BrC chromophores such as nitrophenols in HULIS. Overall, the paper was well-organized, and the results are of broad interest. I would recommend the paper be accepted after revision, as outlined below.*

[Response]

  Thanks for the reviewer's comments on this manuscript. Please check our point-by-point response and the modified text in the manuscript.

*(1) line 48-50. Please elaborate further on the assertion that HULIS is under low-temperature conditions for the majority of its atmospheric lifetime. This statement serves as a critical foundation for the research motivation and requires more evidenced or references to support it.*

[Response]

  We sincerely appreciate the reviewer's insightful comment. In response, we now explicitly link the sharp decrease in ambient temperature during the vertical of transport of aerosols to the atmospheric evolution of HULIS. Furthermore, we have added key citations from (Heald et al., 2005; Liu et al., 2014; Pani et al., 2022; Textor et al., 2006; Wu et al., 2018), all of which provide observational support for the characteristic temperature regimes encountered by aerosols undergoing vertical and long-range transport, thus confirming the prolonged exposure of HULIS to low temperatures during its lifetime in the atmosphere. We believe this revised statement provides a much stronger and evidence-based foundation for the subsequent discussion on the need to understand low-temperature effects on HULIS properties and behavior.

[Revised]

 Once emits into or form in the atmosphere, vertical transport increased the altitude of HULIS-containing particles, leading to long-range transport (Chen et al., 2021; Slade et al., 2017). During vertical transportation, the ambient temperature sharply decreases, indicating that the atmospheric evolution of HULIS accompanies with low-temperature conditions during their majority lifetime (Heald et al., 2005; Liu et al., 2014; Pani et al., 2022; Textor et al., 2006; Wu et al., 2018).

*(2) line 83-84 and Text S1. This semi-quantification strategy was based on the experiment experience that substances with similar structures and equal concentrations yield comparable signal intensities in mass spectrometry. Please specify the criteria for selecting the semi-quantification proxy compounds. Additionally, a thorough discussion on the uncertainties associated with this semi-quantification approach should be provided.*

[Response]

We sincerely appreciate the reviewer's insightful comment. To address the concerns, we have added **Text S2** in the supporting information to clarify the criteria for selecting the semi-quantification proxy compounds and discuss the uncertainties of semi-quantitative strategy. Our selection of proxy compounds is guided by structural similarity to the target analytes, as ionization efficiency in mass spectrometry is dominantly determined by molecular structures – a principle illustrated in prior literature (Ma et al., 2022; Nozière et al., 2015). For instance, vanillin serves as surrogate standard for methoxy- and hydroxy- benzaldehydes; 1-t-butylimidazole quantifies imidazole derivatives; and 4-nitrophenol, 2,6-dimethyl-4-nitrophenol, and 4-methyl-5-nitrocatechol act as proxy compounds for the quantification of nitrophenols derivatives.

**Text S2. Discussion on the uncertainty of semi-quantitative strategy**

In atmospheric chemistry, the components of organic aerosols are always complex and no authentic standards can be used for quantification. In the analysis of these components, it has been widely suggested to use available proxy compounds for quantification (Ma et al., 2022; Nozière et al., 2015). For example, camphor-10-sulfonic acid is often used as surrogate standard for the quantification of $\alpha$-pinene derivative organosulfates (Ma et al., 2022; Nguyen et al., 2014a). This strategy can achieve the quantitative analysis of various compounds in organic aerosols, but there are inevitable uncertainties which mainly come from the ionization efficiency difference between non-authentic standards and target analytes in mass spectrometric analysis (Nozière et al., 2015). Generally, the closer the molecular structures of surrogate standard is to the target analyte, the smaller the difference in ionization efficiency, resulting in the similar signal intensities in mass spectrometry.

**Table S4** lists the deviation observed when different compounds were used as surrogate standards for quantification of each standard in this study. For instance, nitrophenol derivatives (4NP, 2N135T, and 4M5NC) showed quantification errors within 130% when used as surrogate standards for 26D4NP. Tetrahydroquinoline (1234THQ), an N-heterocyclic aromatic hydrocarbon, exhibited an error below 110% when quantifying 1tBI. However, even structurally similar surrogate standards may introduce an error of

a factor over 2. Both P3C and I2C belong to N-heterocyclic aromatic aldehydes, yet using I2C as a surrogate for P3C resulted in a 46% error. When the structurally distinct PAAE was used as a surrogate for P3C, the error reached 600%. Therefore, when adopting semi-quantitative strategies, it is critical to first identify the chemical structure of the target analyte and subsequently select surrogate standards with analogous structural features for quantification.

Regarding uncertainties, we provide an in-depth analysis supported by new validation data (Table S4). This table quantifies errors introduced when different proxy compounds are used for specific standards. For example, structurally close proxies (e.g., nitrophenol derivatives quantifying 26D4NP) typically yield errors below 130%, while even minor structural variations (e.g., I2C vs. P3C, both N-heterocyclic aromatic aldehydes) can cause ~46% deviation. This range aligns with the established uncertainty threshold (factor of ~2) reported for semi-quantitative methods in previous studies (Ma et al., 2022). We thus emphasize that while semi-quantification is essential for complex aerosol components lacking authentic standards, its accuracy is contingent on rigorous proxy selection.

[Revised]

**Text S2. Discussion on the uncertainty of semi-quantitative strategy**

In atmospheric chemistry, the components of organic aerosols are always complex and no authentic standards can be used for quantification. In the analysis of these components, it has been widely suggested to use available proxy compounds for quantification (Ma et al., 2022; Nozière et al., 2015). For example, camphor-10-sulfonic acid is often used as surrogate standard for the quantification of α-pinene derivative organosulfates (Ma et al., 2022; Nguyen et al., 2014a). This strategy can achieve the quantitative analysis of various compounds in organic aerosols, but there are inevitable uncertainties which mainly come from the ionization efficiency difference between non-authentic standards and target analytes in mass spectrometric analysis (Nozière et al., 2015). Generally, the closer the molecular structures of surrogate standard is to the target analyte, the smaller the difference in ionization efficiency, resulting in the similar signal intensities in mass spectrometry.

**Table S4** lists the deviation observed when different compounds were used as surrogate standards for quantification of each standard in this study. For instance, nitrophenol derivatives (4NP, 2N135T, and 4M5NC) showed quantification errors within 130% when used as surrogate standards for 26D4NP. Tetrahydroquinoline (1234THQ), an N-heterocyclic aromatic hydrocarbon, exhibited an error below 110% when quantifying 1tBI. However, even structurally similar surrogate standards may introduce an error of a factor over 2. Both P3C and I2C belong to N-heterocyclic aromatic aldehydes, yet using I2C as a surrogate for P3C resulted in a 46% error. When the structurally distinct PAAE was used as a surrogate for P3C, the error reached 600%. Therefore, when adopting semi-quantitative strategies, it is critical to first identify the chemical structure of the target analyte and subsequently select surrogate standards with analogous structural features for quantification.

**Table S4.** The deviations observed when different compounds were used as surrogate standards for quantifying each target analyte

| Deviation % | E2OCA | ES | SA | F25DC | VAN | 18NaPA | 9FLU | 910PheQ | 2AF | 2FA | 1234THQ | 1tBI | 4NP |
|---|---|---|---|---|---|---|---|---|---|---|---|---|---|
| E2OCA | 100.0 | 4.3 | 8.8 | 17.6 | 101.9 | 64.2 | 9.5 | 324.7 | 21.7 | 2.5 | 163.1 | 155.8 | 93.5 |
| ES | 2400.0 | 100.0 | 207.6 | 421.0 | 2446.7 | 1539.0 | 224.3 | 7799.2 | 519.1 | 57.3 | 3916.1 | 3741.7 | 3014.4 |
| SA | 1164.8 | 47.8 | 100.0 | 203.7 | 1187.5 | 746.6 | 108.1 | 3787.0 | 251.3 | 27.0 | 1901.1 | 1816.4 | 1018.8 |
| F25DC | 570.7 | 23.7 | 49.2 | 100.0 | 581.8 | 365.9 | 53.2 | 1854.7 | 123.3 | 13.5 | 931.2 | 889.7 | 625.2 |
| VAN | 98.1 | 4.8 | 9.1 | 17.8 | 100.0 | 63.2 | 9.8 | 317.2 | 21.8 | 3.0 | 159.6 | 152.6 | 125.5 |
| 18NaPA | 155.5 | 7.3 | 14.2 | 27.9 | 158.5 | 100.0 | 15.3 | 503.5 | 34.3 | 4.5 | 253.2 | 242.0 | 51.2 |
| 9FLU | 1079.1 | 44.1 | 92.5 | 188.5 | 1100.1 | 691.6 | 100.0 | 3508.8 | 232.7 | 24.9 | 1761.4 | 1682.8 | 785.6 |
| 910PheQ | 30.3 | 0.6 | 2.0 | 4.7 | 30.9 | 19.2 | 2.2 | 100.0 | 6.0 | 0.1 | 49.9 | 47.6 | 28.5 |
| 2AF | 463.1 | 19.1 | 39.9 | 81.1 | 472.1 | 296.9 | 43.1 | 1505.4 | 100.0 | 10.9 | 755.8 | 722.1 | 409.9 |
| 2FA | 4219.8 | 175.1 | 364.2 | 739.6 | 4301.9 | 2705.6 | 393.7 | 13714.5 | 912.1 | 100.0 | 6885.9 | 6579.1 | 4191.6 |
| 1234THQ | 61.2 | 2.2 | 5.0 | 10.5 | 62.4 | 39.1 | 5.4 | 199.5 | 13.0 | 1.1 | 100.0 | 95.5 | 60.7 |
| 1tBI | 64.0 | 2.3 | 5.2 | 10.9 | 65.3 | 40.9 | 5.6 | 208.9 | 13.5 | 1.2 | 104.7 | 100.0 | 60.9 |
| 4NP | 106.9 | 3.3 | 9.8 | 16.0 | 79.7 | 195.5 | 12.7 | 351.1 | 24.4 | 2.4 | 164.7 | 164.2 | 100.0 |
| 2N135T | 103.7 | 3.2 | 9.5 | 15.5 | 77.3 | 186.1 | 12.3 | 339.9 | 23.7 | 2.3 | 159.6 | 159.1 | 97.0 |
| 26D4NP | 91.1 | 3.8 | 7.9 | 16.0 | 92.9 | 58.4 | 8.5 | 296.1 | 19.7 | 2.2 | 148.7 | 142.1 | 86.3 |
| 4M5NC | 110.6 | 4.4 | 9.4 | 19.3 | 112.7 | 70.9 | 10.2 | 359.8 | 23.8 | 2.5 | 180.6 | 172.5 | 123.6 |
| 4NA | 131.5 | 5.5 | 11.4 | 23.1 | 134.1 | 84.3 | 12.3 | 427.5 | 28.4 | 3.1 | 214.6 | 205.1 | 184.7 |
| 3NSA | 2423.3 | 100.5 | 209.1 | 424.7 | 2470.4 | 1553.7 | 226.0 | 7875.9 | 523.8 | 57.4 | 3954.4 | 3778.2 | 5453.3 |
| P3C | 91.3 | 4.0 | 8.1 | 16.2 | 93.0 | 58.6 | 8.7 | 296.1 | 19.9 | 2.4 | 148.8 | 142.2 | 92.6 |
| I2C | 200.5 | 8.3 | 17.3 | 35.1 | 204.4 | 128.5 | 18.7 | 651.6 | 43.3 | 4.7 | 327.1 | 312.6 | 209.9 |
| 3PAAE | 15.4 | 0.8 | 1.4 | 2.8 | 15.7 | 9.9 | 1.5 | 49.6 | 3.4 | 0.5 | 25.0 | 23.9 | 14.1 |
| 2AP | 132.3 | 5.5 | 11.4 | 23.2 | 134.9 | 84.8 | 12.3 | 430.1 | 28.6 | 3.1 | 215.9 | 206.3 | 210.9 |
| 4MSPAA | 139.6 | 6.0 | 12.3 | 24.7 | 142.3 | 89.6 | 13.3 | 453.2 | 30.4 | 3.6 | 227.7 | 217.6 | 163.9 |
| 4A5M2MBSA | 1347.0 | 55.3 | 115.7 | 235.6 | 1373.2 | 863.5 | 125.1 | 4379.3 | 290.7 | 31.3 | 2198.5 | 2100.5 | 1400.4 |
| 3M4P1SA | 763.1 | 31.3 | 65.5 | 133.4 | 778.0 | 489.1 | 70.8 | 2481.0 | 164.6 | 17.7 | 1245.5 | 1190.0 | 1021.5 |

| Deviation % | 2N135T | 26D4NP | 4M5NC | 4NA | 3NSA | P3C | I2C | 3PAAE | 2AP | 4MSPAA | 4A5M2MBSA | 3M4P1SA |
|---|---|---|---|---|---|---|---|---|---|---|---|---|
| E2OCA | 96.4 | 109.7 | 90.4 | 76.1 | 4.3 | 109.6 | 50.0 | 654.7 | 75.6 | 71.6 | 7.6 | 13.3 |
| ES | 3108.7 | 2634.1 | 2170.5 | 1824.8 | 99.5 | 2630.1 | 1197.5 | 15728.3 | 1814.0 | 1717.7 | 179.6 | 315.9 |
| SA | 1050.7 | 1278.5 | 1053.3 | 885.5 | 47.5 | 1276.6 | 580.8 | 7637.8 | 880.2 | 833.4 | 86.4 | 152.6 |
| F25DC | 644.8 | 626.3 | 516.1 | 433.9 | 23.5 | 625.4 | 284.7 | 3740.5 | 431.3 | 408.4 | 42.6 | 75.0 |
| VAN | 129.4 | 107.6 | 88.8 | 74.8 | 4.7 | 107.4 | 49.3 | 639.0 | 74.3 | 70.4 | 8.0 | 13.5 |
| 18NaPA | 53.7 | 170.6 | 140.7 | 118.4 | 7.2 | 170.3 | 78.0 | 1014.6 | 117.7 | 111.5 | 12.4 | 21.2 |
| 9FLU | 810.2 | 1184.4 | 975.8 | 820.3 | 43.8 | 1182.7 | 537.9 | 7076.9 | 815.4 | 772.0 | 79.9 | 141.2 |
| 910PheQ | 29.4 | 33.3 | 27.3 | 22.9 | 0.6 | 33.3 | 14.8 | 202.4 | 22.7 | 21.5 | 1.6 | 3.4 |
| 2AF | 422.7 | 508.3 | 418.8 | 352.1 | 19.0 | 507.5 | 230.9 | 3036.0 | 350.0 | 331.4 | 34.5 | 60.8 |
| 2FA | 4322.7 | 4631.4 | 3816.1 | 3208.3 | 174.2 | 4624.5 | 2105.0 | 27658.2 | 3189.2 | 3019.8 | 315.1 | 554.8 |
| 1234THQ | 62.6 | 67.2 | 55.3 | 46.4 | 2.2 | 67.1 | 30.3 | 402.6 | 46.1 | 43.7 | 4.3 | 7.8 |
| 1tBI | 62.8 | 70.3 | 57.8 | 48.6 | 2.3 | 70.2 | 31.7 | 421.6 | 48.3 | 45.7 | 4.4 | 8.1 |
| 4NP | 103.1 | 115.8 | 80.9 | 54.2 | 1.8 | 108.0 | 47.6 | 708.2 | 47.4 | 61.0 | 7.1 | 9.8 |
| 2N135T | 100.0 | 112.3 | 78.4 | 52.5 | 1.8 | 104.7 | 46.2 | 687.0 | 46.0 | 59.2 | 6.9 | 9.5 |
| 26D4NP | 89.0 | 100.0 | 82.4 | 69.3 | 3.8 | 99.8 | 45.4 | 597.2 | 68.9 | 65.2 | 6.8 | 12.0 |
| 4M5NC | 127.5 | 121.4 | 100.0 | 84.0 | 4.4 | 121.2 | 55.1 | 725.7 | 83.5 | 79.1 | 8.1 | 14.4 |
| 4NA | 190.4 | 144.4 | 118.9 | 100.0 | 5.4 | 144.1 | 65.6 | 862.1 | 99.4 | 94.1 | 9.8 | 17.3 |
| 3NSA | 5623.9 | 2659.7 | 2191.5 | 1842.4 | 100.0 | 2655.7 | 1208.8 | 15883.5 | 1831.5 | 1734.2 | 180.9 | 318.6 |
| P3C | 95.5 | 100.2 | 82.6 | 69.4 | 4.0 | 100.0 | 45.6 | 597.0 | 69.0 | 65.4 | 7.0 | 12.2 |
| I2C | 216.5 | 220.0 | 181.3 | 152.4 | 8.3 | 219.7 | 100.0 | 1314.0 | 151.5 | 143.5 | 15.0 | 26.4 |
| 3PAAE | 14.6 | 16.9 | 13.9 | 11.7 | 0.8 | 16.8 | 7.7 | 100.0 | 11.6 | 11.0 | 1.3 | 2.1 |
| 2AP | 217.5 | 145.2 | 119.7 | 100.6 | 5.4 | 145.0 | 66.0 | 867.4 | 100.0 | 94.7 | 9.9 | 17.4 |
| 4MSPAA | 169.0 | 153.2 | 126.3 | 106.2 | 6.0 | 153.0 | 69.8 | 913.8 | 105.6 | 100.0 | 10.7 | 18.6 |
| 4A5M2MBSA | 1444.2 | 1478.5 | 1218.1 | 1024.0 | 55.0 | 1476.3 | 671.6 | 8832.5 | 1017.9 | 963.8 | 100.0 | 176.6 |
| 3M4P1SA | 1053.5 | 837.6 | 690.1 | 580.1 | 31.1 | 836.3 | 380.5 | 5003.8 | 576.6 | 546.0 | 56.6 | 100.0 |

*(3) Move compound identification and quantification from SI to the main text and a discussion of semi-quantification errors, as suggested above, should be incorporated into the manuscript (either in the main text or the SI).*

[Response]

We thank the reviewer for this constructive suggestion. In accordance with the recommendation, compound identification and quantification originally in the supporting information (**Text S1**) have now been moved to Section 2.2 of the main text.

[Revised]

Line 104-120: The obtained data analysis was performed with the Compound Discoverer 3.3 software to generate reasonable molecular formulas and match fine structures to MS/MS data. The numbers of atoms restriction of formula were 1-40 for C, 1-100 for H, 0-40 for O, 0-6 for N, and 0-2 for S, with $0.3 \leq H/C \leq 3.0$, $0 \leq O/C \leq 1.2$, $0 \leq N/C \leq 1.0$, and $0 \leq S/C \leq 0.8$ (Kind and Fiehn, 2007). All of the mathematically formulas for each peak were performed with a mass tolerance of $\pm$ 5 ppm and peak areas more than three times of the blank sample. Three curated spectral databases, mzcloud library database, ChemSpider library database, and CFM-ID (https://cfmid.wishartlab.com) were applied to screen suspect candidates of structure (Allen et al., 2015). According to the Schymanski's confidence level (CL), these candidates were divided into confirmed structures (CL1), probable structures (CL2), and tentative candidates (CL3) (Schymanski et al., 2014). We showed two examples to illustrate the derivation processes of candidates in **Figure S6**.

A semi-quantitative strategy was conducted as follows: target analytes were quantified using external standard solutions of structurally analogous surrogate compounds (Nguyen et al., 2014b; Nozière et al., 2015). A representative application involved utilizing the standard curve of 4-methyl-5-nitrocatechol to simultaneously quantify three structural analogs: 4-methyl-5-nitrocatechol, 3-methyl-5-nitrocatechol, and 3,4-dimethyl-5-nitrocatechol. While this strategy enables quantification of compounds without commercially available standards, it introduces inherent uncertainties due to ionization efficiency variations between surrogates and target analytes (discussed in **Text S2**).

*(4) Section 3.2. In the discussion on the potential sources of HULIS, the authors selected two PM haze events instead of high HULIS episodes. What's HULIS concentration during these two PM pollution events. Additionally, please include the mass spectra of HULIS samples from non-haze days for comparison with those in Event I and II. This would clarify whether the identified sources were specific to HULIS or more generally associated with PM$_{2.5}$.*

[Response]

We appreciate the reviewer's insightful suggestions for strengthening our source analysis. In response, we have now integrated the requested data into Section 3.2. First, the HULIS-C concentrations during both haze events are explicitly stated: 6.68 $\mu$gC m$^{-3}$ for Event I and 4.65 $\mu$gC m$^{-3}$ for Event II, confirming the elevated HULIS loading of two haze events (2-3 times of the average value in non-haze

period, PM$_{2.5}$ concentration < 75 $\mu$g m$^{-3}$). Second, we have added comparative mass spectra of representative clean-day sample in Figure 2C. This comparison demonstrates that the majority of identified compounds are inherent to regional HULIS, while the concentrations of 9-fluorenone, nitrophenols, and other species emitted from coal combustion and biomass burning in the clean-day sample were significantly reduced. These revisions support our conclusion that coal combustion and biomass burning are important sources of HULIS in Northeast China, winter.

[Revised]

Line 169-177: To investigate drivers of the high concentrations and variable light absorption efficiency of HULIS in this study, we selected two samples (Event I and II) among all haze events (PM$_{2.5}$ concentration > 75 $\mu$g m$^{-3}$) that exhibited the maximal divergence in MAE$_{365}$ values. Event I had higher PM$_{2.5}$ (159.6 ± 53.8 $\mu$g m$^{-3}$) and HULIS-C (6.68 $\mu$gC m$^{-3}$) but lower MAE$_{365,HULIS}$ (1.56 m$^2$ gC$^{-1}$), while Even II had lower PM$_{2.5}$ (83.7 ± 36.4 $\mu$g m$^{-3}$) and HULIS-C (4.65 $\mu$gC m$^{-3}$) but higher MAE$_{365,HULIS}$ (2.06 m$^2$ gC$^{-1}$). These contrasting events were chosen for potential sources comparison from the perspective of molecular composition. Considering the lowest PM$_{2.5}$ and HULIS-C concentration, the sample on January 13 (PM$_{2.5}$ = 14.1 ± 11.9 $\mu$g m$^{-3}$, HULIS-C = 0.97 $\mu$gC m$^{-3}$, MAE$_{365,HULIS}$ = 1.28 m$^2$ gC$^{-1}$) was selected to represent clean days. **Figure 2** exhibited the reconstructed MS spectra, the number, and concentration fraction of HULIS in both positive and negative modes.

[Figure]

**Figure 2.** Reconstructed mass spectra of identified HULIS compounds during Event I (A), Event II (B), and Background (C). Spectra are shown for positive ionization mode (left panels) and negative ionization mode (right panels). m/z values increase from middle to both sides in all spectra. The most abundant ions are labeled with their m/z values. The accompanying pie charts represent the molecular class distribution

of the identified compounds: the inner/outer ring shows the relative abundance based on number/concentration of compounds.

(5) *Section 3.3. The discussion of the effect of temperature on BrC chromophore formation is arguably the most important aspect of this study. As the author proposed, low temperature promoted the exothermic process (e.g., reaction of phenols with NOx radicals) and hindered the endothermic chemical reactions (e.g., atmospheric oxidation of nitrophenols), thus facilitating the accumulation of BrC chromophores in HULIS. In addition to this qualitative thermodynamic explanation, a more in-depth discussion is encouraged. Specifically, how does low temperature alter the overall atmospheric chemistry and also any comparable results/findings/ mechanistic insights from previous literature that support your conclusions.*

[Response]

We deeply appreciate the reviewer's recognition of this study's core contribution and call for deeper mechanistic insights. While systematic experimental data on BrC evolution under low temperature remain limited, we have enhanced the discussion from the perspective of particle-phase retention mechanism and field observation evidence. Specifically, low temperature suppresses volatilization and enhances particle-phase retention of semi-volatile chromophores (He et al., 2006; Huang et al., 2006), providing a physicochemical pathway for BrC accumulation. This is directly corroborated by field observations of elevated nitroaromatic compound abundance in winter aerosols (Cai et al., 2022; Teich et al., 2017; Zhang et al., 2024), with researchers specifically linking such enhancement to "lower ambient temperatures in winter" (Cai et al., 2022; Zhang et al., 2024). These additions bridge our thermodynamic framework with empirical evidence, demonstrating how temperature alters atmospheric chemistry through both molecular stabilization and inhibition of degradation pathways. The mechanistic understanding of cold-region BrC chemistry requires more systematic experimental simulations to verify.

[Revised]

Line 247-260: Secondly, the formation of BrC chromophores was also important for the $MAE_{365}$ enhancement of HULIS. On the one hand, the secondary formation of nitrophenols has been conclusively attributed to reaction of phenols with $NO_x$ radicals (Bolzacchini et al., 2001; Finewax et al., 2018; Kroflič et al., 2021; Mayorga et al., 2021), a process that has been characterized as exothermic (Bolzacchini et al., 2001; Domingo et al., 2021). On the other hand, we have demonstrated that further atmospheric oxidation of nitrophenols proceeds via a ring-opening mechanism of benzene moiety (Qiu et al., 2024), which constitutes an endothermic reaction (Cao et al., 2021; Hems and Abbatt, 2018; Wang et al., 2017). From a thermodynamic perspective, low temperature not only promotes exothermic formation of nitrophenols while simultaneously suppressing their endothermic degradation via ring-opening. Furthermore, low temperature inhibits the volatilization and enhances the particle-phase retention of these volatile chromophores (He et al., 2006; Huang et al., 2006). This combined effect of low temperature led to the accumulation of strong BrC chromophores like nitrophenols within HULIS. This mechanism is consistent with field observations of enhanced nitroaromatic abundance in winter aerosols (Cai et al., 2022; Teich et al., 2017; Zhang et al., 2024). As such, we infer that ambient temperature plays

[revised manuscript text omitted]